# Citrin mediated metabolic rewiring in response to altered basal subcellular Ca$^{2+}$ homeostasis

Zhanat Koshenov [1], Furkan E. Oflaz[1], Martin Hirtl[1], Benjamin Gottschalk [1], Rene Rost[1], Roland Malli [1,2] &
Wolfgang F. Graier [1,2 ✉]

In contrast to long-term metabolic reprogramming, metabolic rewiring represents an instant and reversible cellular adaptation to physiological or pathological stress. Ca$^{2+}$ signals of distinct spatio-temporal patterns control a plethora of signaling processes and can determine basal cellular metabolic setting, however, Ca$^{2+}$ signals that define metabolic rewiring have not been conclusively identified and characterized. Here, we reveal the existence of a basal Ca$^{2+}$ flux originating from extracellular space and delivered to mitochondria by Ca$^{2+}$ leakage from inositol triphosphate receptors in mitochondria-associated membranes. This Ca$^{2+}$ flux primes mitochondrial metabolism by maintaining glycolysis and keeping mitochondria energized for ATP production. We identified citrin, a well-defined Ca$^{2+}$-binding component of malate-aspartate shuttle in the mitochondrial intermembrane space, as predominant target of this basal Ca$^{2+}$ regulation. Our data emphasize that any manipulation of this ubiquitous Ca$^{2+}$ system has the potency to initiate metabolic rewiring as an instant and reversible cellular adaptation to physiological or pathological stress.

[1] Molecular Biology and Biochemistry, Gottfried Schatz Research Center, Medical University of Graz, Neue Stiftingtalstraße 6/6, 8010 Graz, Austria.
[2] BioTechMed Graz, 8010 Graz, Austria. ✉email: wolfgang.graier@medunigraz.at

Metabolic rewiring represents a hallmark in cellular adaptations to physiological (e.g. anaerobic glycolysis upon lack of oxygen during exercise)[1] and pathological stress (e.g. metabolic overflow or cancer)[2–4]. To achieve an instant and reversible adaptation, such processes particularly require minimal effort to yield maximal effect when affecting basic settings of cellular metabolism. It is tempting to speculate that $Ca^{2+}$, a ubiquitous second messenger with a plethora of effector proteins[5], is involved in such control of basal metabolic setting of a cell. Notably, for a cell's survival and wellbeing, the cellular $Ca^{2+}$ homeostasis has to function correctly despite all kinds of physiological and pathological challenges the cell may face[6]. Mitochondria, the powerhouses of cells, are known to be tightly regulated by $Ca^{2+}$ ions[7,8], and understanding the basic regulation processes of mitochondrial bioenergetics represents an important task considering the fundamental role of this organelle in the cell's energy metabolism and involvements in multiple signaling cascades[9].

The foremost source of $Ca^{2+}$ for mitochondria is the endoplasmic reticulum (ER)[10]. Upon cell stimulation with an inositol 1,4,5-trisphosphate- ($IP_3$-) generating agonist, the transfer of $Ca^{2+}$ from ER to mitochondria occurs predominantly in specialized regions called Mitochondria Associated ER membranes (MAMs)[10]. Thereby $Ca^{2+}$ passes the outer mitochondrial membrane (OMM) through voltage-dependent anion channels (VDACs)[11] into the mitochondrial inter-membrane space (IMS), where it needs to surpass a certain threshold to finally be taken up to mitochondrial matrix by mitochondrial calcium uniporter complex (MCUC)[12–15]. This flux of $Ca^{2+}$ from ER to mitochondria is known to regulate vital processes in the mitochondrial matrix, including the activity of $Ca^{2+}$ sensitive mitochondrial dehydrogenases[16].

Alteration in ER-mitochondria communication is reported to be involved in numerous pathological conditions, including ER stress[17], age-related diseases, such as cancer[18] and neurodegeneration[19], and metabolic and cardiovascular disorders[20]. During early stages of ER stress, cells develop a small ER $Ca^{2+}$ leak that is sensed by mitochondria, which boosts mitochondrial ATP production to ensure proper energy support counteracting hampered ER protein folding[17,21]. Progression of ER stress triggers significant $Ca^{2+}$ loss from the ER yielding long-term mitochondrial $Ca^{2+}$ overload resulting in mitochondrial dysfunction and, eventually, the initiation of apoptotic cell death[22]. Accordingly, ER Unfolded Protein Response (UPR) during aging is associated with enforced ER-mitochondria tethering[23,24], which may eventually result in mitochondrial $Ca^{2+}$ overload and, thus, can explain aging-associated oxidative stress[23] and apoptotic cell death[25]. Notably, mitochondrial bioenergetics and regulation of mitochondrial $Ca^{2+}$ uptake are also greatly changed in cancer[26,27], where alterations in mitochondrial $Ca^{2+}$ signaling play a role in cancer resistance to chemotherapy and increased metastasis[28,29]. Hence, in many metabolic and neurodegenerative diseases, mitochondrial $Ca^{2+}$ signaling and ER-mitochondria crosstalk are altered, resulting in changes in mitochondrial bioenergetics that add to the disease progression and outcome[20], pointing to fundamental importance of ER to mitochondrion $Ca^{2+}$ crosstalk and its alterations in cellular physiology and pathology.

Although it is widely assumed that the mitochondrial matrix residing $Ca^{2+}$-sensitive dehydrogenases are the end-point receivers of changes in mitochondrial $Ca^{2+}$ signaling, an in-depth analysis of the contribution of basal $Ca^{2+}$ homeostasis on mitochondrial bioenergetics and other possible players being involved in the putative $Ca^{2+}$ control of the cell's resting metabolic settings has not been performed so far. In order to understand how basal $Ca^{2+}$ homeostasis affects mitochondrial wellbeing, especially regarding their main attribute, energy production, we sought to study the regulation of basal mitochondrial bioenergetics by resting $Ca^{2+}$ homeostasis. A clear understanding of this fundamental process of basal $Ca^{2+}$ regulation of mitochondrial bioenergetics will help us to tackle many pathologies involving altered mitochondrial $Ca^{2+}$ signaling and inter-organellar communication and would shed light on metabolic rewiring capability of resting $Ca^{2+}$ homeostasis.

We have adapted a simple model to address this question that enabled us to slightly reduce basal $Ca^{2+}$ level in the cytosol, ER, IMS, and matrix. Consequently, we distinguished between two main mechanisms of $Ca^{2+}$ regulation of basal mitochondrial bioenergetics, the mitochondrial surface $Ca^{2+}$-controlled citrin in the IMS, and the MCU-dependent matrix $Ca^{2+}$-sensitive dehydrogenases. We also studied $Ca^{2+}$ dependent mitochondrial pyruvate utilization and explored the importance of MAM vs. cytosolic $Ca^{2+}$ for the regulation of basal mitochondrial bioenergetics. Notably, we have shown that a small decrease in IMS $Ca^{2+}$ level strongly affects the cell's metabolism by switching it to a state of pseudo hypoxia, whereupon the cell relies less on mitochondria and more on glycolysis for energy production despite the presence of oxygen.

## Results

**Short-term removal of extracellular $Ca^{2+}$ controls mitochondrial bioenergetics despite minor effects on matrix $Ca^{2+}$ level**. To establish a protocol that minimally perturbs basal subcellular $Ca^{2+}$ homeostasis affecting the mitochondria and submitochondrial compartments, we implemented a seemingly easy and widely used method of removing extracellular $Ca^{2+}$. Removal of extracellular $Ca^{2+}$ yielded a very small drop in basal cytosolic $Ca^{2+}$ level (Fig. 1a, i) that was accompanied with similar attenuations in the ER (Fig. 1a, ii), IMS (Fig. 1, iii) and the mitochondrial matrix (Fig. 1a, iv) within the first 5–7 min (Supplementary Fig. 1a–i) in cells of the human endothelial cell line EA.hy926.

Short-term removal of extracellular $Ca^{2+}$ did not change the number of ER-mitochondria contacts sites (MAMs) (Fig. 1b, c), and that of ER-Plasma Membrane (PM) contacts (Fig. 1d, e, left panel). However, removal of extracellular $Ca^{2+}$ decreased the average size of ER-PM contact sites (Fig. 1e, middle panel) and the total area of ER in the PM proximity (Fig. 1e, right panel).

In order to reveal the effect of disturbance in the cell's basal $Ca^{2+}$ homeostasis on mitochondrial bioenergetics, we tested the main attribute the mitochondria are known for, the ATP synthesis. Mitochondrial ATP producing capability was assessed by administering ATP synthase inhibitor oligomycin while measuring mitochondrial ATP level ($[ATP]_{mito}$) using genetically encoded ATP sensor mtAT1.03[30]. The drop in $[ATP]_{mito}$ upon the administration of oligomycin reflects the actual ATP production by mitochondria (Fig. 2a, c). Disturbing basal mitochondrial $Ca^{2+}$ level by the means of extracellular $Ca^{2+}$ removal prior to oligomycin addition resulted in an increase of $[ATP]_{mito}$ in response to oligomycin in app. 35% of the cells (Fig. 2b, c). This increase in mitochondrial ATP in response to oligomycin indicates that in one-third of all cells tested in the absence of extracellular $Ca^{2+}$ the ATP synthase switched its direction and started to consume ATP instead of producing it. In the remaining two-thirds of the cells perfused with $Ca^{2+}$ free buffer, mitochondrial ATP production was significantly diminished (Fig. 2c). The reversal of mitochondrial ATP synthase and reduction of ATP production in response to $Ca^{2+}$ imbalance were rescued by returning to nutrient supplemented $Ca^{2+}$ containing buffer for 10 min after $Ca^{2+}$ removal (Fig. 2c), pointing to a dynamic regulation and versatility of this $Ca^{2+}$ dependent phenomenon. In almost all of the measured cells, switching to

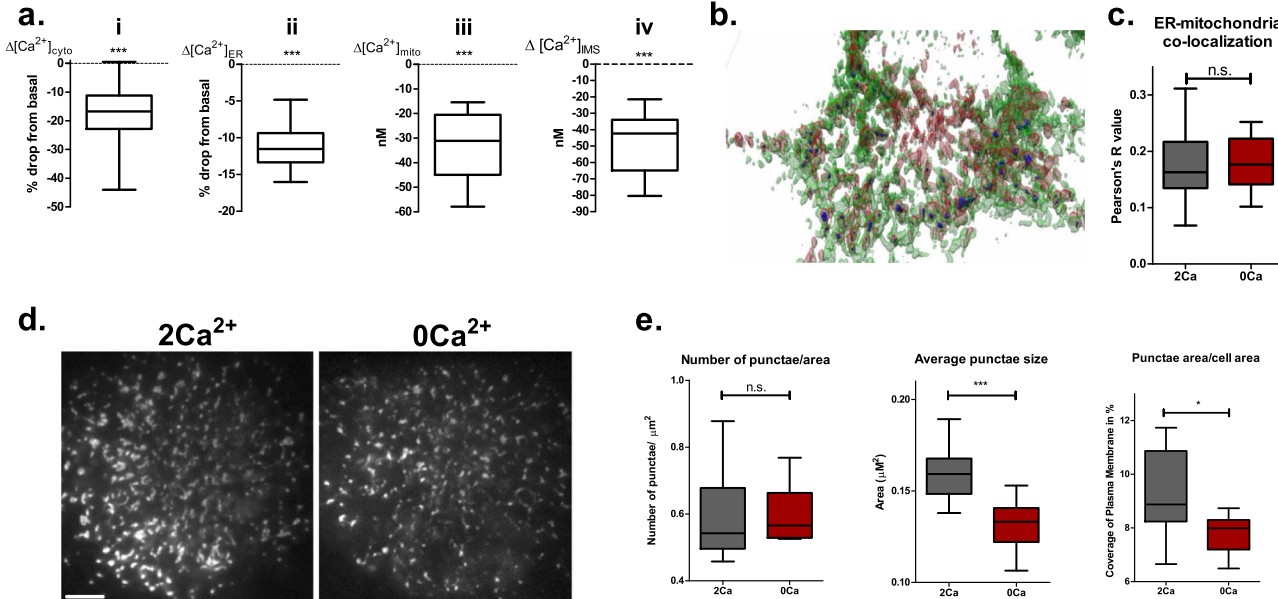

**Fig. 1 Removal of extracellular $Ca^{2+}$ triggers basal $Ca^{2+}$ drop in intracellular compartments and affects ER-PM contact cites. a** Removal of extracellular $Ca^{2+}$ reduces basal $Ca^{2+}$ levels in cytosol ($n = 64$), ER ($n = 17$), IMS ($n = 13$), and mitochondria ($n = 21$). **b** Exemplary 3D reconstruction of EA.hy926 cell with pseudo- colored ER (green) and mitochondria (red) used for analysis of ER-mitochondria contacts (dark blue). **c** Analysis of ER-mitochondria contacts ($n = 22$, unpaired $t$-test; n.s.; $p = 0.68$). **d** Exemplary TIRF images of ER in the plasma membrane proximity in 2 $Ca^{2+}$ (left) and 0 $Ca^{2+}$ (right) conditions, scale bar (left bottom corner) is 5 µm. **e** Statistical analysis of ER-PM contacts (unpaired $t$-test, $n = 9$; n.s. $p = 0.92$,; *$p = 0.018$; ***$p = 0.004$).

$Ca^{2+}$ free buffer is followed by a slow reduction of $[ATP]_{mito}$, suggesting that the reduced activity of the ATP synthase precedes its reversal (Fig. 2b, d).

Simultaneous measurement of mitochondrial membrane potential ($\Psi_{mito}$) and $[ATP]_{mito}$ allowed us to monitor the contribution of the reversed mode of ATP synthase to maintaining $\Psi_{mito}$ under the condition of reduced basal $Ca^{2+}$. In cells with intact basal subcellular $Ca^{2+}$ homeostasis, $\Psi_{mito}$ remained stable even upon the addition of oligomycin (Fig. 2a). In contrast, in cells with perturbed basal subcellular $Ca^{2+}$ homeostasis (i.e., in the absence of extracellular $Ca^{2+}$), oligomycin initiated a strong and continuous decrease in $\Psi_{mito}$ (Fig. 2b).

To clarify the effect of the perturbed basal $Ca^{2+}$ homeostasis achieved by removal of extracellular $Ca^{2+}$ on mitochondrial bioenergetics, mitochondrial NADH production, which is regulated by $Ca^{2+}$[31,32] was measured. While basal NADH levels were comparable, maximum NADH production was strongly reduced under condition of perturbed basal $Ca^{2+}$ homeostasis (Fig. 2e, f). This points to insufficient mitochondrial NADH production and inability to fuel electron transport chain (ETC) in the absence of extracellular $Ca^{2+}$. Hence, the reduced ETC activity also provides a potential reason for the reduced ATP production and reversal of ATP synthase, whereupon ATP synthase is reversed in order to prevent the total collapse of $\Psi_{mito}$. Similar to mitochondrial ATP, NADH production in cells with disturbed subcellular $Ca^{2+}$ homeostasis can be rescued by $Ca^{2+}$ re-addition (Supplementary Fig. 2a, b).

To further test our assumptions, mitochondrial respiration was measured. In the absence of extracellular $Ca^{2+}$, basal, maximal, and ATP-coupled oxygen consumption rates (OCR) were reduced in cell populations (Fig. 2g, h), thus, confirming the results obtained with single-cell ATP and NADH measurements mentioned above. Accordingly, our data indicate that besides the fundamental role of $Ca^{2+}$ as a crucial mediator during cell stimulation, basal subcellular $Ca^{2+}$ homeostasis greatly controls resting mitochondrial bioenergetics. Additionally, the findings

point to a pivotal role of extracellular $Ca^{2+}$ in maintaining basal subcellular $Ca^{2+}$ homeostasis.

**Acute perturbation of subcellular $Ca^{2+}$ homeostasis only minimally affects mitochondrial NADH production by $Ca^{2+}$ sensitive dehydrogenases.** To test whether the drop in basal matrix $Ca^{2+}$ upon transient removal of extracellular $Ca^{2+}$ and the subsequent reduction in basal activity of matrix dehydrogenases are responsible for altered mitochondrial metabolism, we performed NADH measurements in the presence of pyruvate (1 mM) to overcome potential substrate limitations due to subcellular $Ca^{2+}$ misbalance. Pyruvate supplementation resulted in elevated basal NADH level (Fig. 3a) and showed no difference in NADH production between control and 0 $Ca^{2+}$ conditions (Fig. 3b, c), thus, indicating that the diminished NADH production upon removal of extracellular $Ca^{2+}$ was unlikely to be due to reduced mitochondrial dehydrogenase activity. To further test this assumption, the activity of pyruvate dehydrogenase (PDH), which is known to be indirectly regulated by $Ca^{2+}$[33,34], was assessed by measuring mitochondrial citrate production under extracellularly added pyruvate (1 mM) using the mitochondrial-targeted genetically encoded citrate sensor CitrON[35]. Mitochondrial PDH activity was not affected by alterations in the basal subcellular $Ca^{2+}$ homeostasis as indicated by the identical increase in mitochondrial citrate upon pyruvate administration under control and absence of extracellular $Ca^{2+}$ conditions (Fig. 3d, e). Additionally, by measuring mitochondrial pyruvate with matrix targeted PyronicSF[36], we showed that mitochondrial pyruvate uptake was also not affected by acute $Ca^{2+}$ deprivation (Fig. 3f, g). But, as it is known that pyruvate itself, as well as NADH and ATP/ADP ratio, can affect PDH phosphorylation and thus its activity[37], we assessed the phosphorylation status of PDH under our experimental conditions (Fig. 3h, i, Supplementary fig. 3). Short-term removal of extracellular $Ca^{2+}$ only slightly increased the phosphorylation of PDH,

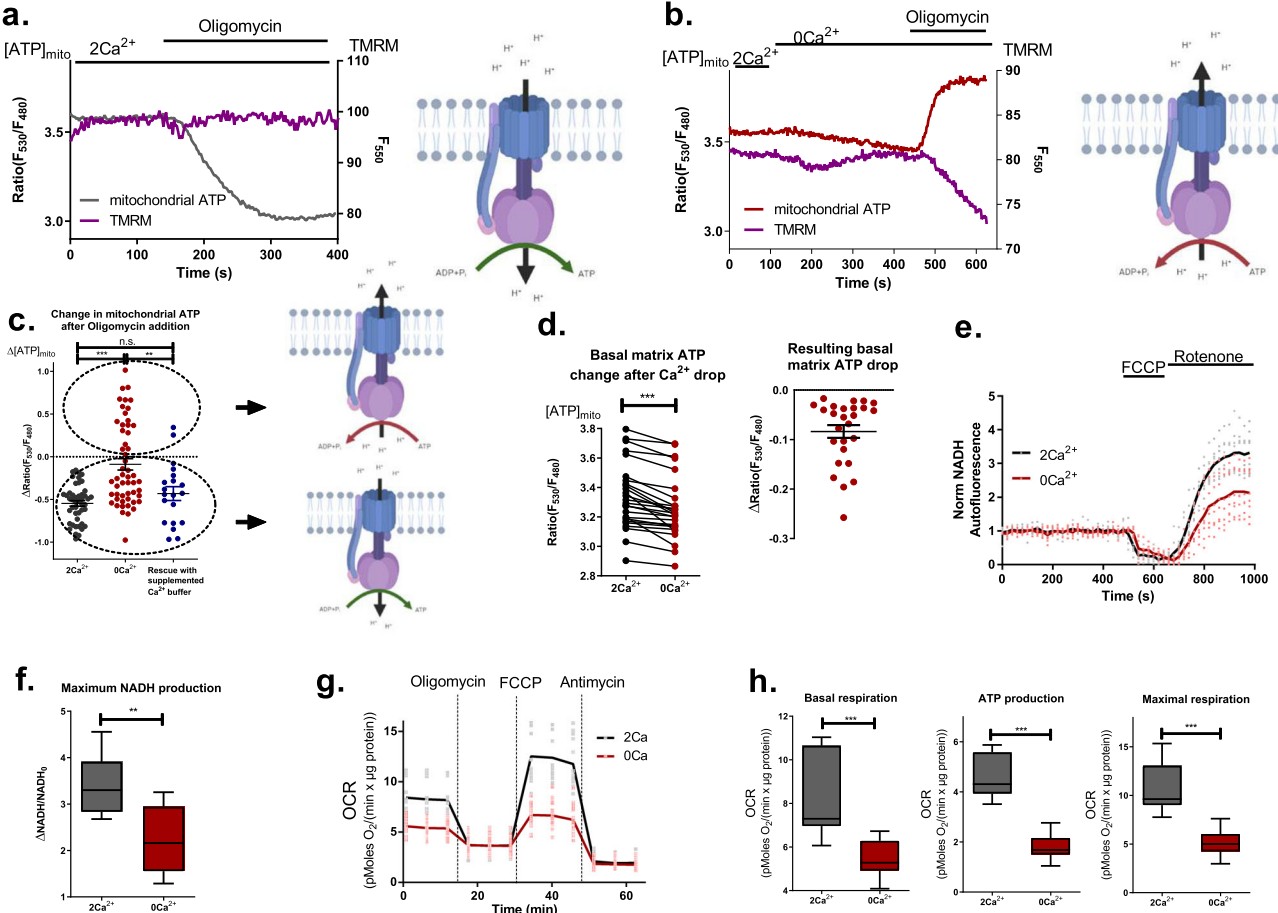

**Fig. 2 Perturbance of basal Ca$^{2+}$ homeostasis alters mitochondrial bioenergetics. a** Representative control traces of simultaneous measurements of [ATP]$_{mito}$ and $\Psi_{mito}$ using mtAT1.03 and TMRM, respectively. On the right, a graphical illustration of the ATP synthase mode is depicted (Created with BioRender.com.). **b** Representative traces of [ATP]$_{mito}$ and $\Psi_{mito}$ in 0 extracellular Ca$^{2+}$. Graphical representation of the ATP synthase mode is depicted on the right (Created with BioRender.com.). **c** Statistical analysis of the change in [ATP]$_{mito}$ after 2 µM oligomycin with graphical illustration of ATP synthase modes on the right (Created with BioRender.com.) (One-way ANOVA with Tukey's multiple comparison test, $n = 46$ for 2 Ca$^{2+}$, $n = 53$ for 0 Ca$^{2+}$, $n = 20$ for the rescue experiment; ***$p < 0.001$, **$p < 0.01$). **d** Statistical analysis of the change in [ATP]$_{mito}$ after 5 min in 0 Ca$^{2+}$ buffer (on the left) (paired $t$-test, $n = 26$; ***$p < 0.0001$). **e** Average traces (solid lines) and single data points of mitochondrial NADH autofluorescence measurements. **f** Respective statistical analysis of maximal NADH production (unpaired $t$-test, $n = 8$; **$p = 0.004$). **g** Average traces (solid lines) and single data points of OCR measurements. **h** Respective statistical analysis of OCR data from **g** (unpaired $t$-test, $n = 17$ for 2 Ca$^{2+}$, $n = 25$ for 0 Ca$^{2+}$; ***$p < 0.0001$).

with much greater increase after 1 h (Fig. 3h, i), indicating that there might be an additional effect of short-term removal of extracellular Ca$^{2+}$ beside its influence on PDH phosphorylation.

**Perturbance in basal subcellular Ca$^{2+}$ homeostasis induces metabolic rewiring.** Interestingly, while OCR data did not clarify the mechanism of how perturbation in intracellular Ca$^{2+}$ homeostasis affects mitochondrial bioenergetics despite the differences observed (Fig. 2g), the elevation of extracellular acidification rate (ECAR) upon the removal of extracellular Ca$^{2+}$ (Fig. 4a) points at a possible increased lactate production and, thus, metabolic rewiring caused by the disturbed subcellular Ca$^{2+}$ homeostasis.

To further explore the observed metabolic changes upon perturbations in basal subcellular Ca$^{2+}$ homeostasis, cytosolic lactate level was assessed. Notably, removal of extracellular Ca$^{2+}$ yielded instant accumulation of cytosolic lactate (Fig. 4b, c), which is in line with increased ECAR (Fig. 4a). To understand if this lactate accumulation was metabolically relevant, we measured lactate under glucose-free conditions and observed that lactate accumulation in response to Ca$^{2+}$ removal was gone in the absence of glucose (Fig. 4d). Glucose utilization and uptake were

not affected by the removal of Ca$^{2+}$ (Fig. 4e). Because cellular lactate level is tightly correlated with cytosolic NADH/NAD$^+$ balance, we tested the cytosolic redox state in order to better understand the mechanism behind lactate accumulation. By measuring cytosolic NADH/NAD$^+$ ratio with genetically encoded sensor Peredox[38] we observed an increase in the NADH/NAD$^+$ ratio under the condition of disturbed subcellular Ca$^{2+}$ homeostasis (Fig. 4f–h). This observation provides a possible link between redox misbalance and lactate accumulation, since upon increase of cytosolic NADH/NAD$^+$ ratio lactate dehydrogenase (LDH) may convert pyruvate to lactate in order to restore the redox balance, which, in turn, would result in increased lactate production[39] and metabolic rewiring. Excitingly, the metabolic rewiring achieved by perturbed basal subcellular Ca$^{2+}$ homeostasis was reversible by Ca$^{2+}$ re-addition (Supplementary Fig. 4a–c).

**Citrin is responsible for cytosolic and mitochondrial metabolic rewiring during disturbed subcellular Ca$^{2+}$ homeostasis.** Among several Ca$^{2+}$-sensitive proteins involved in cytosolic and mitochondrial redox balance and metabolism, the calcium-binding mitochondrial carrier protein citrin, which resides in

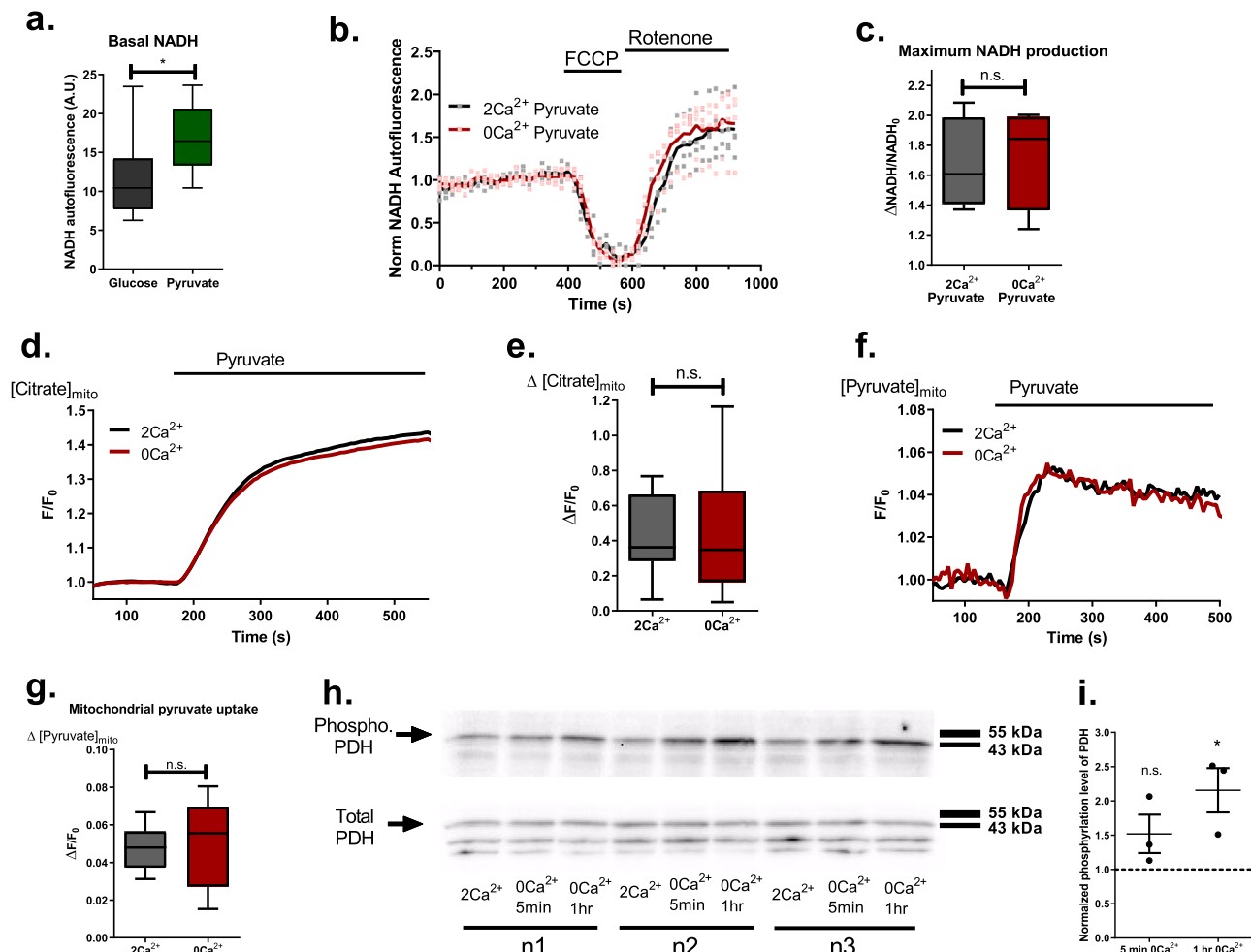

**Fig. 3 Mitochondrial pyruvate uptake and utilization are not affected under reduced basal Ca$^{2+}$ condition. a** Basal mitochondrial NADH autofluorescence of cells in either glucose or glucose and pyruvate supplemented buffers (unpaired $t$-test, $n = 16$ for glucose, $n = 8$ for pyruvate supplementation, *$p = 0.0168$). **b** Average traces (solid lines) and single data points of mitochondrial NADH autofluorescence measurements with pyruvate supplementation. **c** Statistical analysis of **b** (unpaired $t$-test, $n = 4$; n.s., $p = 0.788$). **d** Representative traces of mitochondrial citrate measurements with mito-CitrON. **e** Statistical analysis of mitochondrial citrate production measurements as shown in **d** (unpaired $t$-test, $n = 21$; n.s., $p = 0.875$). **f** Representative traces of mitochondrial pyruvate measurements with mito-PyronicSF. **g** Statistical analysis of mitochondrial pyruvate uptake as shown in **f** (unpaired $t$-test, $n = 8$; n.s., $p = 0.819$). **h** Western blot analysis of phosphorylation status of PDH; cells were incubated in 2 Ca$^{2+}$, 0 Ca$^{2+}$ for 5 min and 0 Ca$^{2+}$ for 1 hr, then harvested on ice and blotted for phosphorylated PDH at Ser293 (upper blot); total PDH was blotted after mild stripping of phospho. PDH antibody (lower blot). **i** Statistical analysis of **h**, total PDH was used as loading control; horizontal dotted line represents control level (Repeated Measures ANOVA with Tukey's multiple comparison test, $n = 3$, *$p < 0.05$).

the mitochondrial inter-membrane space (IMS) and serves as part of the malate-aspartate shuttle (MAS)[40,41] is a good candidate for the role of a translator of the disturbed subcellular Ca$^{2+}$ homeostasis to cellular and mitochondrial metabolic setting in our model. To elucidate whether the metabolic changes upon disturbed subcellular Ca$^{2+}$ homeostasis were due to perturbation of IMS Ca$^{2+}$-controlled citrin activity, we measured cytosolic lactate (Fig. 5a, b), NADH/NAD$^+$ ratio (Fig. 5c, d), mitochondrial NADH (Fig. 5e–g) and ATP levels (Fig. 6a, b) under transient knockdown of citrin (Supplementary Fig. 5 a–d). Basal lactate level (Fig. 5a) and NADH/NAD$^+$ ratio (Fig. 5c) were elevated in citrin-depleted cells. Notably, the increase of lactate (Fig. 4b) and NADH/NAD$^+$ ratio (Fig. 4e, f) in response to perturbation of subcellular Ca$^{2+}$ homeostasis were both prevented by knockdown of citrin (Fig. 5b, d). Hence, the knockdown of citrin reduced mitochondrial basal NADH level and maximum NADH production (Fig. 5e–g) as well as basal mitochondrial ATP level (Fig. 6a). These results indicate that the disturbance of subcellular Ca$^{2+}$ homeostasis upon removal of extracellular Ca$^{2+}$ and

subsequently, the decrease in IMS Ca$^{2+}$ (Fig. 1a), mimics the effects of citrin knockdown. Importantly, citrin KD didn't have a drastic effect on cell viability and apoptosis in the timeframe of our experiments (Supplementary Fig. 6a, b).

**Differential regulation of mitochondrial bioenergetics by IMS Ca$^{2+}$ dependent citrin and MCU dependent matrix Ca$^{2+}$ and dehydrogenases.** In order to verify the individual importance of IMS Ca$^{2+}$ and Citrin versus matrix Ca$^{2+}$ and dehydrogenases in regulation of resting mitochondrial bioenergetics, we compared the effect of citrin knockdown with that of MCU (Supplementary Fig. 5a–d). Basal mitochondrial ATP level was reduced under both citrin and MCU knockdown conditions, while the reduction was more pronounced in citrin-depleted cells (Fig. 6a). In line with these findings, the shift of ATP synthase towards its reverse mode and reduced ATP production in control cells with perturbed subcellular Ca$^{2+}$ homeostasis was comparable with that found in citrin-depleted cells in both normal and 0 Ca$^{2+}$

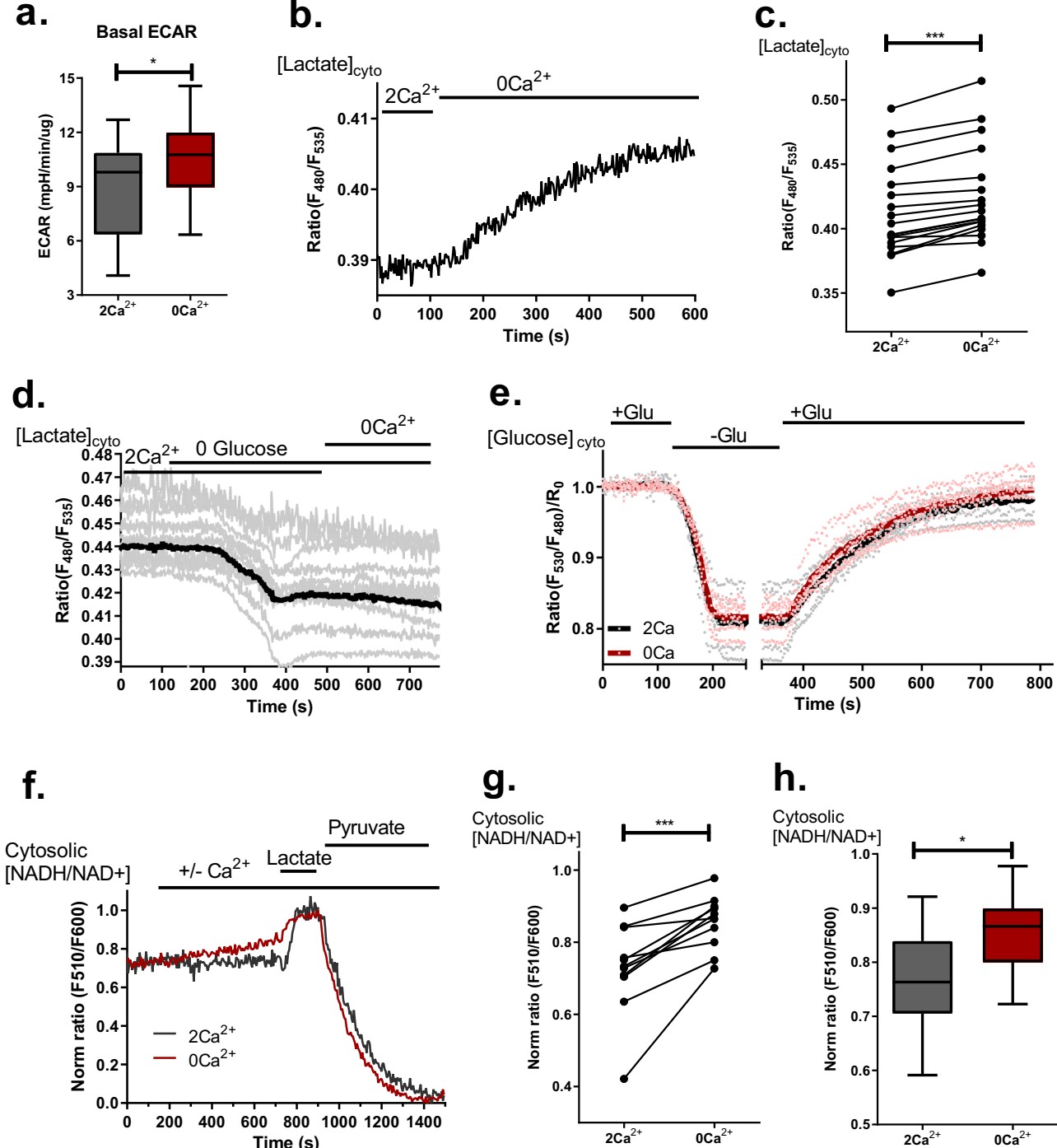

**Fig. 4 ECAR, cytosolic lactate and NADH/NAD$^+$ ratio increase in response to Ca$^{2+}$ deprivation. a** Statistical analysis of ECAR (unpaired $t$-test, $n = 17$ for 2 Ca$^{2+}$, $n = 25$ for 0 Ca$^{2+}$; *$p = 0.0414$). **b** Representative trace of cytosolic lactate accumulation measured with Laconic after perfusion with 0 Ca$^{2+}$ buffer. **c** Statistical analysis of cytosolic lactate levels before and after 6 min of Ca$^{2+}$ removal (paired $t$-test, $n = 18$; ***$p < 0.0001$). **d** Average trace (solid black line) and single cell traces (grey lines) of cytosolic lactate measurements with glucose removal ($n = 12$). **e** Average traces (solid lines) and single data points of cytosolic glucose measurements using FLII12Pglu-700μδ6 with glucose removal and re-addition ($n = 5$ for 2 Ca$^{2+}$, $n = 6$ for 0 Ca$^{2+}$). **f** Representative traces of cytosolic NADH/NAD$^+$ measurements with Peredox and subsequent calibration with 10 mM lactate and pyruvate. **g** Statistical analysis of cytosolic NADH/NAD$^+$ ratio of cells before and after 6 min of Ca$^{2+}$ removal (paired $t$-test, $n = 11$; ***$p = 0.0003$). **h** Comparison of cytosolic NADH/NAD$^+$ ratio after 6 min in 2 Ca$^{2+}$ or 0 Ca$^{2+}$ buffer (unpaired $t$-test, $n = 9$ for 2 Ca$^{2+}$, $n = 11$ for 0 Ca$^{2+}$; *$p = 0.0258$).

conditions (Fig. 6b). In contrast to citrin knockdown, MCU depletion showed a less drastic reversal of ATP synthase and ATP production under normal Ca$^{2+}$ condition, while a clear additive effect of Ca$^{2+}$ removal was present (Fig. 6b). Similar results were obtained in HeLa cells, pointing to uniformity of the findings

(Supplementary Fig. 7a–c). OCR measurements further corroborated these results as citrin KD significantly reduced ATP production linked respiration, while MCU KD did not (Supplementary Fig. 8a–c). In addition, citrin KD resulted in increased ECAR (Supplementary Fig. 8d), thus supporting the findings

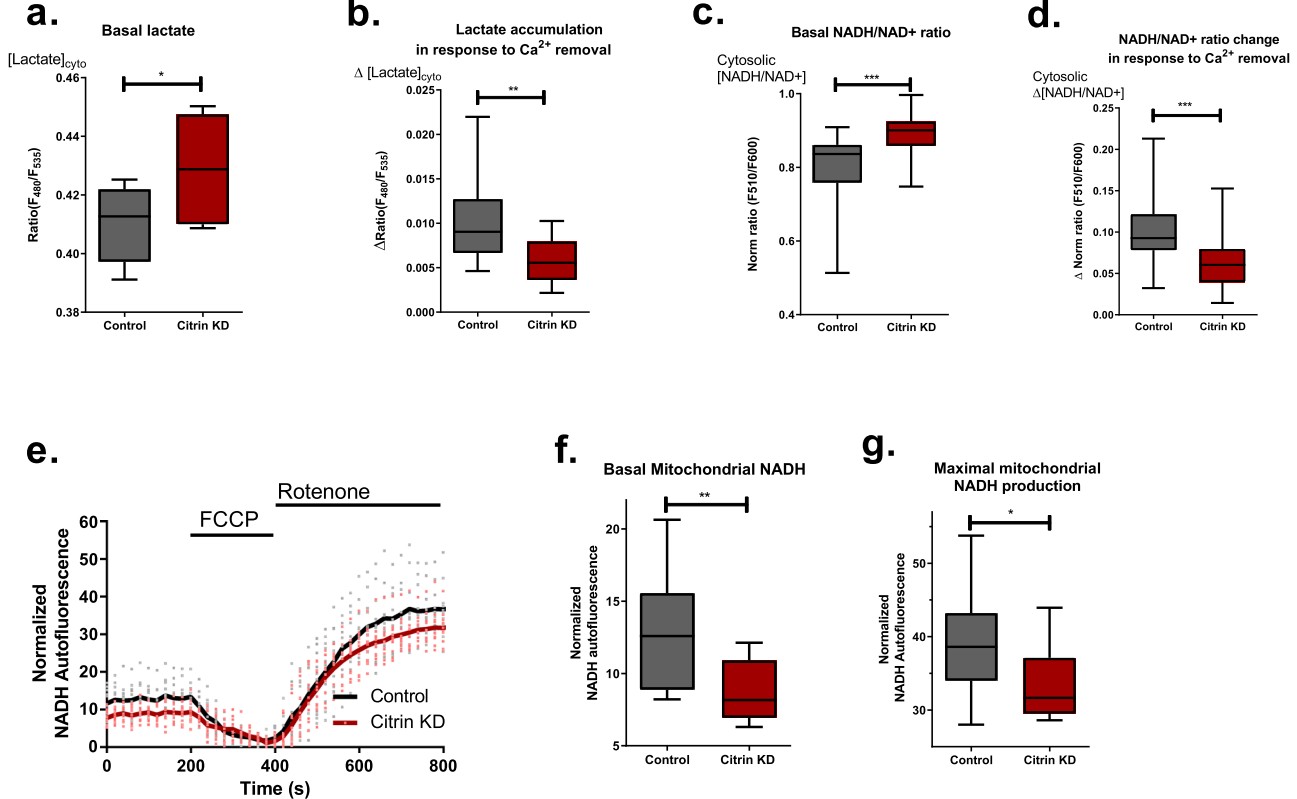

**Fig. 5 Citrin KD mimics the metabolic alterations happening during basal Ca$^{2+}$ drop condition. a** Basal cytosolic lactate levels of control vs Citrin KD in 2 Ca$^{2+}$ (unpaired $t$-test, $n = 7$; *$p = 0.0411$). **b** Lactate accumulation after extracellular Ca$^{2+}$ removal (as in Fig. 4 a) in control vs Citrin KD (unpaired $t$-test, $n = 15$ for control, $n = 13$ for citrin KD; **$p = 0.0064$). **c** Basal cytosolic NADH/NAD$^+$ ratio of control vs Citrin KD in 2 Ca$^{2+}$ (unpaired $t$-test, $n = 30$ for control, $n = 18$ for citrin KD; ***$p = 0.0005$). **d** NADH/NAD$^+$ ratio change after extracellular Ca$^{2+}$ removal (as in Fig. 4 e) in control vs Citrin KD (unpaired $t$-test, $n = 29$ for control, $n = 18$ for citrin KD; **$p = 0.0002$). **e** Average traces (solid lines) and single data points of mitochondrial NADH autofluorescence measurements in control and Citrin KD cells. **f** Statistical comparison of basal mitochondrial NADH in control vs Citrin KD shown in panel **e** (unpaired $t$-test, $n = 11$; **$p = 0.0064$). **g** Maximal NADH production in control vs Citrin KD cells shown in panel **e** (unpaired $t =$ test, $n = 11$; *$p = 0.0333$).

performed with the removal of extracellular Ca$^{2+}$ (Fig. 4a–c). The absence of a more pronounced difference in basal and maximal OCR for citrin KD compared to controls is likely stemming from low transfection efficiency in EA.hy926 cells (Supplementary Fig. 5a–d), which is especially problematic for the kind of experiments that lack positive transfection marker as we have recently shown[21].

Furthermore, we tested whether MCU over expression (OE) or MICU1 KD, which can increase mitochondrial Ca$^{2+}$ uptake[13] or basal matrix Ca$^{2+}$ level[42], respectively, will rescue the metabolic defects of citrin KD. Neither MCU OE nor MICU1 KD could rescue reduced basal mitochondrial ATP level or ATP production of citrin KD cells (Fig. 6c, d), emphasizing once more the importance of citrin and IMS Ca$^{2+}$ for regulation of basal mitochondrial bioenergetics. Despite the dominating role of citrin over MCU for mitochondrial bioenergetics under resting conditions, both proteins share the fundamental importance on histamine-stimulated mitochondrial ATP production (Fig. 6e, f), even though citrin KD doesn't affect basal mitochondrial Ca$^{2+}$ or mitochondrial Ca$^{2+}$ uptake upon stimulation (Supplementary Fig. 9a, b).

**Citrin controls pyruvate availability for mitochondria.** Because the cytosolic NADH recycling function of MAS is involved in regulation of pyruvate production by providing NAD$^+$ for glycolysis, we further analyzed the impact of citrin knockdown and IMS Ca$^{2+}$ imbalance on mitochondrial pyruvate supply by measuring cytosolic pyruvate. Since the sensitivity of the genetically encoded pyruvate sensor Pyronic[43] used in this study did not allow measurements of small pyruvate fluctuations[43] (Fig. 7a), we used inhibitors of the monocarboxylate transporters (MCT) and lactate dehydrogenase (LDH), AR-C155858 and GSK 2837808 A, respectively, to facilitate cytosolic pyruvate accumulation (Fig. 7b). The amount of cytosolic pyruvate accumulation after inhibition of MCT and LDH represents the flux of pyruvate to these respective proteins. While basal cytosolic pyruvate levels were similar in control cells and cells with disturbed basal subcellular Ca$^{2+}$ homeostasis (Fig. 7a), pyruvate accumulation is increased under reduced Ca$^{2+}$ condition and in citrin-depleted cells (Fig. 7b). Interestingly, there was no further difference in pyruvate accumulation between normal and reduced Ca$^{2+}$ conditions in citrin-depleted cells (Fig. 7b). These findings indicate that LDH and MCT use more pyruvate during IMS Ca$^{2+}$ imbalance or in the absence of citrin, thus diminishing pyruvate availability for mitochondria.

To further test this hypothesis we have measured mitochondrial pyruvate directly when going from glucose-free buffer to glucose containing one, thus estimating the mitochondrial pyruvate supply. We have compared control, citrin KD and MCU KD conditions in the presence (Fig. 7c) and absence of extracellular Ca$^{2+}$ (Fig. 7d), and confirmed that citrin and IMS Ca$^{2+}$ control pyruvate supply of mitochondria, as both citrin KD and removal of extracellular Ca$^{2+}$ reduced mitochondrial pyruvate supply (Fig. 7c–e). Importantly, MCU doesn't seem to be involved in this process as MCU KD didn't affect

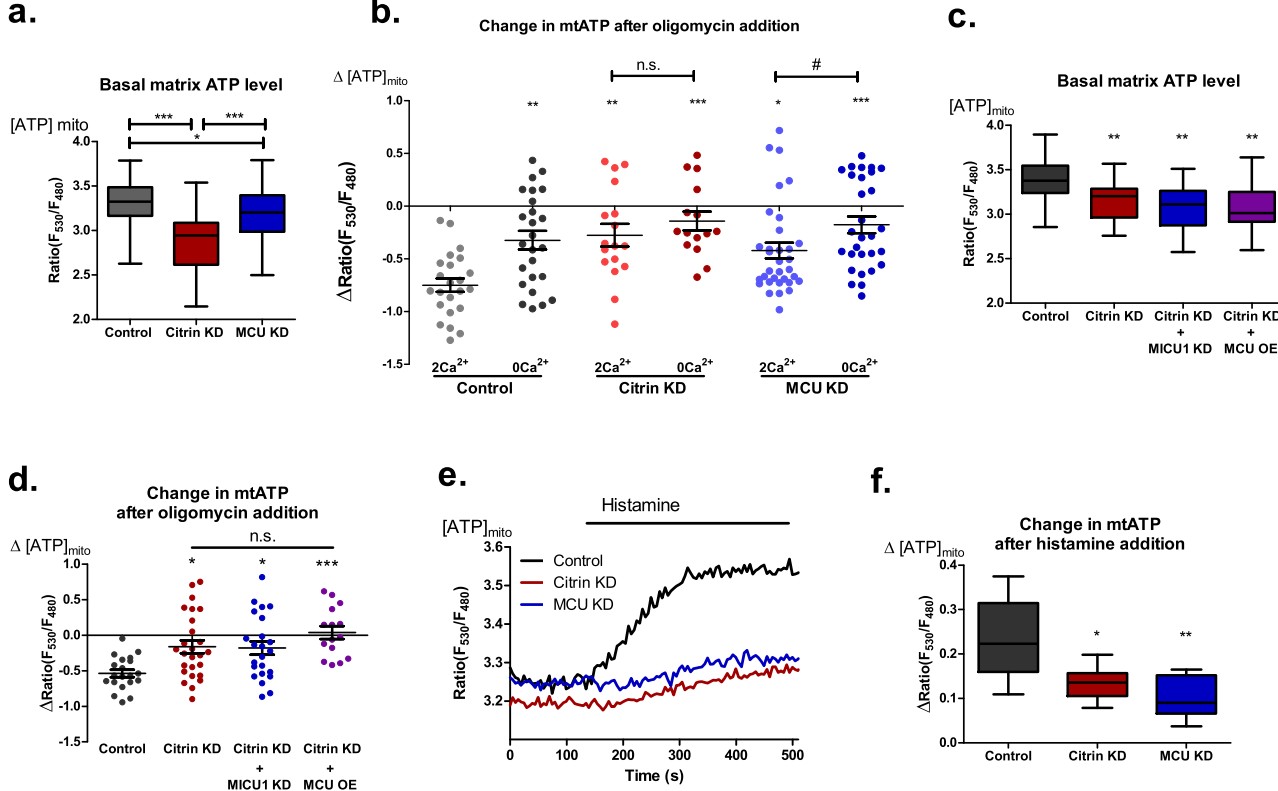

**Fig. 6 Citrin/IMS Ca$^{2+}$ are more important for basal mitochondrial bioenergetics than MCU/matrix Ca$^{2+}$. a** Statistical analysis of basal mitochondrial ATP levels represented by mtAT1.03 ratio in control, MCU KD and citrin KD (One-way ANOVA with Tukey's multiple comparison test, *$p < 0.05$, ***$p < 0.001$, $n = 57$ for control, $n = 32$ for citrin KD, $n = 62$ for MCU KD). **b** Statistical analysis of the change in mitochondrial ATP level after 2 μM oligomycin addition in control, MCU KD and citrin KD cells +/− extracellular Ca$^{2+}$ (One-way ANOVA with Tukey's multiple comparison test, $n = 23$ (control 2 Ca$^{2+}$), 25 (control 0 Ca$^{2+}$), 17 (citrin KD 2 Ca$^{2+}$), 15 (citrin KD 0 Ca$^{2+}$), 33 (MCU KD 2 Ca$^{2+}$), 29 (MCU KD 0 Ca$^{2+}$), *$p < 0.05$, **$p < 0.01$, ***$p < 0.001$; unpaired $t$-test, citrin KD 2 Ca vs citrin KD 0 Ca, n.s. $p = 0.3519$; unpaired $t$-test, MCU KD 2 Ca vs MCU KD 0 Ca, # $p = 0.0286$). **c** Statistical analysis of basal mitochondrial ATP levels represented by mtAT1.03 ratio in control, citrin KD, citrin and MICU1 double KD, and citrin KD MCU OE cells (One-way ANOVA with Tukey's multiple comparison test, **$p < 0.01$, $n = 20$ for control, $n = 26$ for citrin KD, $n = 23$ for citrin and MICU1 double KD, $n = 15$ for citrin KD MCU OE). **d** Statistical analysis of the change in mitochondrial ATP level after 2 μM oligomycin addition in control, citrin KD, citrin and MICU1 double KD, and citrin KD MCU OE cells (One-way ANOVA with Tukey's multiple comparison test, *$p < 0.05$, ***$p < 0.001$, n.s. $p > 0.05$, $n = 20$ for control, $n = 26$ for citrin KD, $n = 23$ for citrin and MICU1 double KD, $n = 15$ for citrin KD MCU OE). **e** Representative traces of mitochondrial ATP measurements upon histamine stimulation in control, citrin KD and MCU KD cells. **f** Statistical analysis of histamine induced mitochondrial ATP production as shown in **e** (One-way ANOVA with Tukey's multiple comparison test, *$p < 0.05$, **$p < 0.01$, $n = 6$ for control, $n = 8$ for citrin KD, $n = 7$ for MCU KD).

mitochondrial pyruvate supply in the presence of extracellular Ca$^{2+}$, while perturbing basal subcellular homeostasis in MCU KD cells did (Fig. 7c–e). Additionally, MCU OE didn't rescue the reduced mitochondrial pyruvate supply of citrin KD cells (Supplementary Fig. 10).

Less pyruvate availability for mitochondria would explain worsened mitochondrial bioenergetics under these conditions. To test this assumption, we measured mitochondrial ATP production under normal and 0 Ca$^{2+}$ conditions, but this time, we supplemented experimental buffers with 1 mM pyruvate. Pyruvate supplementation rescued the effect of IMS Ca$^{2+}$ imbalance on mitochondrial ATP production (Fig. 7f), supporting the hypothesis that reduced pyruvate availability for mitochondria is one of the reasons for worsened mitochondrial bioenergetics when subcellular/IMS Ca$^{2+}$ homeostasis is disturbed.

**MAM, but not cytosolic, Ca$^{2+}$ flux determines basal mitochondrial bioenergetics.** Since the ER represents the main source for mitochondrial Ca$^{2+}$ that is transferred predominantly via the MAMs, we wanted to understand how exactly the removal of extracellular Ca$^{2+}$ affects mitochondrial bioenergetics. Cytosol

and the MAMs are the two routes through which Ca$^{2+}$ might be engaged in the regulation of basal mitochondrial energetics. Theoretically, Ca$^{2+}$ from both these sources could influence IMS Ca$^{2+}$ homeostasis as there is no apparent threshold at the OMM for Ca$^{2+}$ exchange. To address this question, we deployed the intracellular Ca$^{2+}$ chelating agents BAPTA-AM and EGTA-AM. Notably, BAPTA has faster Ca$^{2+}$ binding and is known to be able to buffer spatial Ca$^{2+}$ in e.g. the MAM region[44], whereas EGTA is a rather slow Ca$^{2+}$ chelator and, thus, its effectiveness is limited to buffering global cytosolic Ca$^{2+}$ fluctuations[44]. Basal mitochondrial ATP, as well as mitochondrial ATP production, were greatly reduced when the cells were treated with BAPTA-AM but not EGTA-AM, pointing to the importance of basal MAM-IMS Ca$^{2+}$ homeostasis for regulation of citrin and, subsequently, mitochondrial bioenergetics (Fig. 8a, b). To further elaborate which proteins of the Ca$^{2+}$ toolkit[5] are actually involved in the basal Ca$^{2+}$ homeostasis within the MAM/IMS, we assessed the effect of the IP$_3$R inhibitor Xestospongin C[45], store-operated Ca$^{2+}$ entry (SOCE) inhibitor Pyrozole 6 (Pyr6)[46] and short transient receptor potential channel 3 (TRPC3) inhibitor Pyrozole 10 (Pyr10)[46]. Xestospongin C, but not Pyr6 or Pyr10, reduced mitochondrial ATP production and reversed the ATP synthase

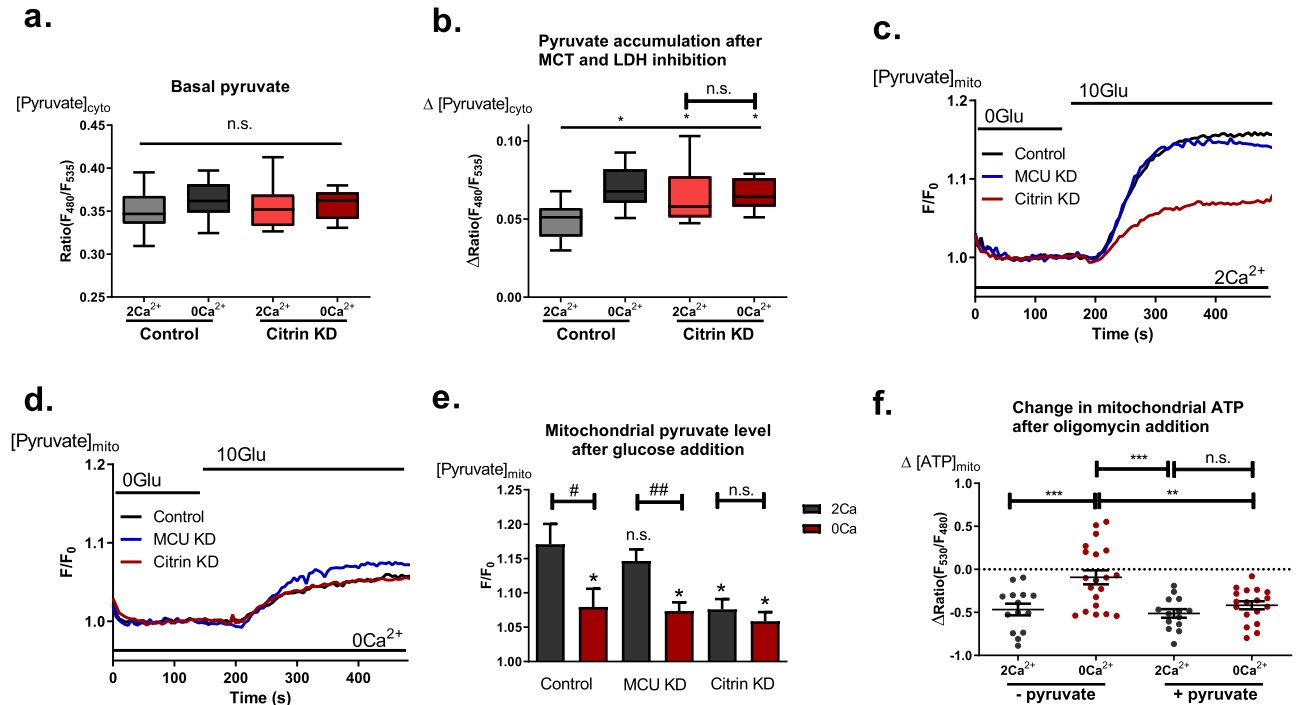

**Fig. 7 Citrin activity regulates pyruvate's fate. a** Basal cytosolic pyruvate levels measured with Pyronic in control vs. knockdown of citrin in the presence and absence of extracellular $Ca^{2+}$ (One-Way ANOVA with Tukey's multiple comparison test, $n = 9$ for control 2 $Ca^{2+}$ and 0 $Ca^{2+}$, $n = 10$ for citrin KD 2 $Ca^{2+}$ and 0 $Ca^{2+}$; **b** Cytosolic pyruvate accumulation in response to LDH and MCT inhibition with GSK 2837808 A and AR-C155858, respectively (One-Way ANOVA with Tukey's multiple comparison test, *$p < 0.05$, $n = 9$ for control 2 $Ca^{2+}$ and 0 $Ca^{2+}$, $n = 10$ for citrin KD 2 $Ca^{2+}$ and 0 $Ca^{2+}$). **c** Representative traces of mitochondrial pyruvate measurements with mito-PyronicSF in the presence of extracellular $Ca^{2+}$ under control, citrin KD and MCU KD conditions. **d** Representative traces of mitochondrial pyruvate measurements with mito-PyronicSF in the absence of extracellular $Ca^{2+}$ under control, citrin KD and MCU KD conditions. **e** Statistical analysis of mitochondrial pyruvate measurements as shown in **c** and **d** (One-Way ANOVA with Tukey's multiple comparison test, $n = 11$ for control 2 $Ca^{2+}$, $n = 10$ for control 0 $Ca^{2+}$, $n = 12$ for MCU KD 2 $Ca^{2+}$, $n = 8$ for MCU KD 0 $Ca^{2+}$, $n = 14$ for citrin KD 2 $Ca^{2+}$, $n = 7$ for citrin KD 0 $Ca^{2+}$, *$p < 0.05$, n.s. $p > 0.05$; unpaired $t$-test, #$p < 0.034$, ##$p = 0.006$, n.s., $p = 0.463$). **f** Comparison of mitochondrial ATP dynamics after 2 μM oligomycin addition in the presence and absence of extracellular $Ca^{2+}$ ±1 mM pyruvate supplementation (One-Way ANOVA with Tukey's multiple comparison test, **$p < 0.01$, ***$p < 0.001$, $n = 14$ for 2 $Ca^{2+}$ no pyruvate), 20 for 0 $Ca^{2+}$ no pyruvate, 14 for 2 $Ca^{2+}$ pyruvate, 18 for 0 $Ca^{2+}$ pyruvate).

direction (Fig. 8c), supporting the importance of IP$_3$Rs in MAM/IMS $Ca^{2+}$ flux even under basal conditions. Interestingly, the impact of combined application of Pyr6 and Pyr10 was similar to that of Xestospongin C (Fig. 8c) and 0 $Ca^{2+}$ (Fig. 2c), likely due to the need of basal plasma membrane $Ca^{2+}$ flux for continuous ER refilling and maintenance of MAM-IMS $Ca^{2+}$ homeostasis.

To validate the results obtained with pharmacological inhibition, we have performed similar experiments with knockdown of IP$_3$R2 and Orai1 as the main proteins responsible for MAM[47] and SOCE[48] pathways of $Ca^{2+}$ delivery to IMS, respectively (Fig. 8d, e). KD of both genes showed a reduction of basal mitochondrial ATP level and ATP production, with IP$_3$R2 KD having a much stronger effect on mitochondrial bioenergetics than Orai1 KD (Fig. 8d, e). These results support the observations made with the pharmacological tools and emphasize the importance of MAM-IMS $Ca^{2+}$ flux as the regulating mechanism of basal mitochondrial bioenergetic wiring.

## Discussion

This work was designed to investigate principal regulating mechanisms of basal mitochondrial bioenergetics as fundamental processes for conditioning cellular metabolic wiring. To achieve this goal, we have deployed a common protocol of extracellular $Ca^{2+}$ removal that allowed us to distinguish differential regulation of mitochondrial metabolism by basal subcellular $Ca^{2+}$ homeostasis. Short-term removal of extracellular $Ca^{2+}$ yields

slight and reversible drops in basal $Ca^{2+}$ levels of multiple organelles and sub-organellar compartments (Fig. 1a, Supplementary Fig. 1). Initially, our prediction was that mitochondrial bioenergetics will worsen as the result of basal drop in matrix $Ca^{2+}$ due to diminished activity of mitochondrial $Ca^{2+}$-sensitive dehydrogenases (Fig. 2e, f). However, the diminished NADH production by mitochondria under reduced basal $Ca^{2+}$ condition was rescued by 1 mM pyruvate supplementation (Fig. 3a, c). This observation cast a doubt on our prediction that this slight drop in basal $Ca^{2+}$ in mitochondrial matrix was the reason for the worsened mitochondrial bioenergetics under this condition. Since the reported $K_D$s of matrix dehydrogenases for $Ca^{2+}$ ions in live cells are in the high nM–μM range[16], possibly the observed acute reduction of 20–40 nM in the matrix was not affecting them. Subsequent direct measurements of mitochondrial pyruvate uptake capability and pyruvate dehydrogenase activity revealed that both are not affected in the given timeframe by the resulting 15% drop of basal $Ca^{2+}$ level (Fig. 3d–g). In line with these functional experiments, the PDH phosphorylation was only slightly increased by short-term removal of extracellular $Ca^{2+}$ (Fig. 3h, i). Collectively, these results indicate that disrupted basal intracellular $Ca^{2+}$ homeostasis achieved by short-term removal of extracellular $Ca^{2+}$ can affect something other than or in addition to mitochondrial dehydrogenases.

The increased ECAR in Seahorse® experiments in the absence of extracellular $Ca^{2+}$ pointed to an increase in glycolytic activity under this condition (Fig. 4a). In line with this finding, lactate

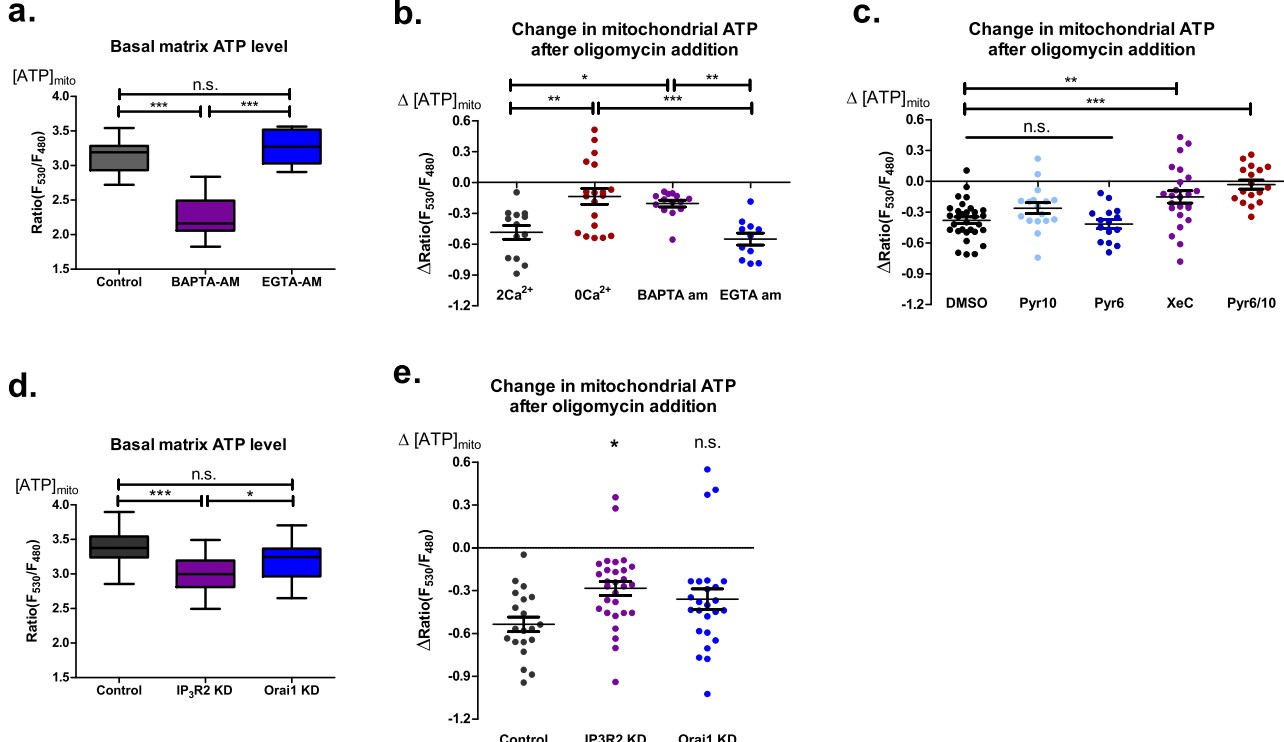

**Fig. 8 Mitochondrial bioenergetics are influenced by basal MAM Ca$^{2+}$ flux. a** Basal matrix ATP levels in control cells and cells incubated with 50 μM BAPTA-AM or 50 μM EGTA-AM for 30 min (One-Way ANOVA with Tukey's multiple comparison test, ***$p < 0.001$, $n = 32$ for control, $n = 14$ for BAPTA-AM, $n = 11$ for EGTA-AM). **b** Analysis of mitochondrial ATP change after 2 μM oligomycin addition to cells perfused with 2 Ca$^{2+}$ or 0 Ca$^{2+}$, and pre-incubated with 50 μM BAPTA-AM or 50 μM EGTA-AM (One-Way ANOVA with Tukey's multiple comparison test, *$p < 0.05$, **$p < 0.01$, ***$p < 0.001$, $n = 13$ for 2 Ca$^{2+}$, $n = 19$ for 0 Ca$^{2+}$, $n = 14$ for BAPTA-AM, $n = 11$ for EGTA-AM). **c** Analysis of mitochondrial ATP change after 2 μM oligomycin addition to cells incubated with DMSO control ($n = 31$), 3 μM Pyr6 ($n = 17$), 3 μM Pyr10 ($n = 15$), 10 μM XeC ($n = 24$), and 3 μM Pyr6/3 μM Pyr10 ($n = 17$), One-Way ANOVA with Tukey's multiple comparison test, **$p < 0.01$, ***$p < 0.001$. **d** Basal matrix ATP levels in control, IP$_3$R2 KD and Orai1 KD cells (One-Way ANOVA with Tukey's multiple comparison test, ***$p < 0.001$, *$p < 0.05$, $n = 20$ for control, $n = 29$ for IP$_3$R2 KD, $n = 25$ for Orai1 KD). **e** Analysis of mitochondrial ATP change after 2 μM oligomycin addition to control, IP$_3$R2 KD and Orai1 KD cells (One-Way ANOVA with Tukey's multiple comparison test, *$p < 0.05$ versus control, $n = 20$ for control, $n = 29$ for IP$_3$R2 KD, $n = 25$ for Orai1 KD).

accumulates in cells with disturbed subcellular Ca$^{2+}$ homeostasis (Fig. 4b, c). Notably, as the lactate accumulation does not occur in the absence of glucose (Fig. 4d) and neither glucose utilization nor glucose import are affected by the removal of extracellular Ca$^{2+}$ (Fig. 4e), this lactate accumulation under disturbed sub-cellular Ca$^{2+}$ homeostasis appears to be physiological. The increase in the cytosolic NADH/NAD$^+$ ratio (Fig. 4f–h) upon perturbation in basal subcellular Ca$^{2+}$ homeostasis led us to speculate that lactate accumulates as a result of increased LDH activity that is enhanced in order to support glycolysis under the condition of increased cytosolic NADH/NAD$^+$ ratio[39].

In line with the assumptions above, we suspected that a Ca$^{2+}$-sensitive process, which is responsible for cytosolic NADH/NAD$^+$ ratio misbalance, affects mitochondrial bioenergetics, and is involved in the metabolic rewiring of mitochondria under the condition of disturbed subcellular Ca$^{2+}$ homeostasis achieved in our Ca$^{2+}$ removal protocol. Accordingly, we propose the Ca$^{2+}$-sensitive mitochondrial solute carrier citrin (SLC25A13)[40,41,49–54] as a potential candidate responsible for Ca$^{2+}$-dependent meta-bolic rewiring of mitochondria in our model. Notably, citrin knockdown mimicked the Ca$^{2+}$ removal phenotype in all of the measured parameters, including increased cytosolic lactate, NADH/NAD$^+$ ratio, reduced mitochondrial NADH and ATP production (Figs. 5, 6a, b), as well as increased pyruvate con-sumption by LDH (Fig. 7b) and reduced mitochondrial pyruvate supply (Fig. 7c–e). These data support the role of MAS in reg-ulation of basal mitochondrial bioenergetics and in linking

cytosolic and mitochondrial metabolism via mitochondrial sur-face/IMS Ca$^{2+}$ (Fig. 9). Hence, any disruption of the IMS Ca$^{2+}$ homeostasis results in the rewiring of metabolism, as IMS Ca$^{2+}$ regulated citrin activity fuels the matrix with NADH equivalents and pyruvate (Fig. 7a–e) for the TCA cycle that generates electron donors (Fig. 2e, f) for ETC, making even a slight reduction in IMS Ca$^{2+}$ (Fig. 1a) a big impact for mitochondrial bioenergetics.

Additionally, the comparison of the knockdown of citrin with that of MCU on mitochondrial ATP production (Fig. 6) and pyruvate supply (Fig. 7c–e) supports the concept of the impor-tance of MAM-IMS Ca$^{2+}$ modulation of mitochondrial bioener-getics via citrin but not MCU under resting conditions. In addition to its importance under resting conditions, our data indicate that citrin is as important as MCU for boosting mito-chondrial bioenergetics upon stimulation (Fig. 6e, f). The impor-tance of IMS Ca$^{2+}$ for regulation of mitochondrial bioenergetics and basal metabolic wiring is further supported by our findings that neither MCU OE nor MICU1 KD can rescue defective mitochondrial bioenergetics resulting from citrin KD (Fig. 6c, d, Supplementary Fig. 10a, b). These findings highlight the impor-tance of MAS and its regulation by MAM-IMS Ca$^{2+}$ flux under resting and stimulated conditions and, thus, Ca$^{2+}$-activated citrin (and MAS) essentially need to be considered besides Ca$^{2+}$-sensi-tive matrix dehydrogenases when evaluating Ca$^{2+}$-activated pro-cesses that control mitochondrial bioenergetics.

In a recent study it has been shown that pyruvate utilization by mitochondria is Ca$^{2+}$ dependent[55]. Notably, mitochondrial

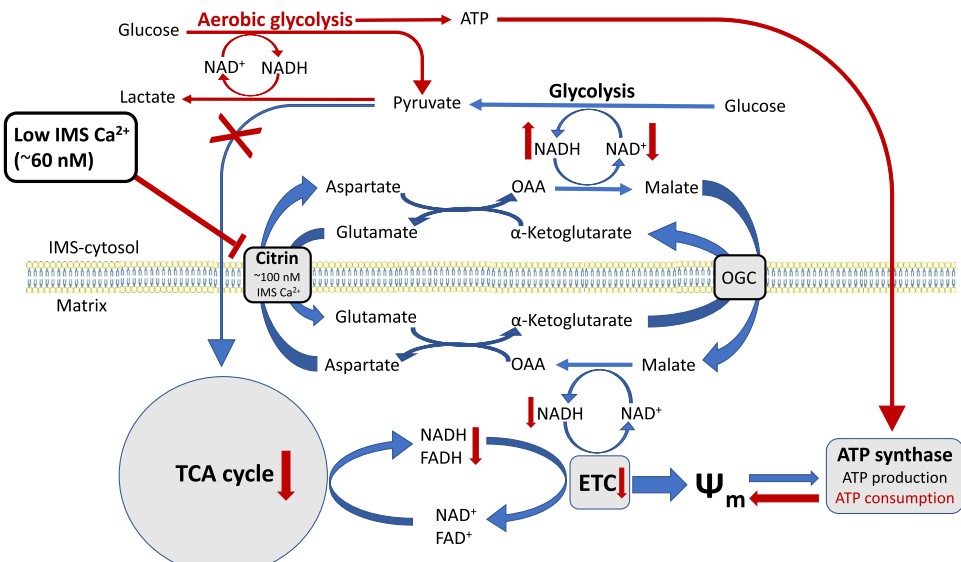

**Fig. 9 Schematic representation of IMS $Ca^{2+}$ homeostasis as fundamental regulator of resting cytosolic and mitochondrial metabolism.** IMS $Ca^{2+}$ modulates citrin/malate-aspartate shuttle activity with basal IMS $Ca^{2+}$ level being set at around 100 nM by ER-mitochondrial $Ca^{2+}$ flux. Main processes that are regulated by basal IMS $Ca^{2+}$ driven MAS activity are shown with blue arrows, and consist of MAS recycling $NAD^+$ for glycolysis and glycolytic products going through TCA cycle and ETC. Disruption of this $Ca^{2+}$ homeostasis leads to metabolic rewiring, shown here with red arrows and text. As a result, cells switch to aerobic glycolysis, rely on LDH for NADH recycling, and have reduced OxPhos. OAA, oxaloacetate; OGC, oxoglutarate/malate carrier protein.

pyruvate metabolism has two $Ca^{2+}$ regulation phases. First, direct, $Ca^{2+}$-dependent regulation of pyruvate utilization by mitochondrial matrix dehydrogenases, and second, indirect, $Ca^{2+}$-dependent regulation of pyruvate availability for mitochondria by citrin/MAS activity. Since in addition to direct substrate supply to mitochondria, citrin, in a $Ca^{2+}$-sensitive manner, recycles cytosolic NADH and makes $NAD^+$ available for glycolysis, it controls production and fate of pyruvate as shown here and by others[52,56], while not affecting the mitochondrial pyruvate transport itself (Fig. 3f, g). When the cytosolic NADH/$NAD^+$ ratio is increased as a result of basal MAM/IMS $Ca^{2+}$ disruption and reduced citrin activity, pyruvate gets converted to lactate by LDH with a concurrent conversion of NADH to $NAD^+$, thus reestablishing the balanced redox ratio of NAD to sustain glycolysis. In support of this view, a recent study pointed to the need for $NAD^+$ in driving aerobic glycolysis[57]. But as a consequence of this metabolic rewiring, mitochondria get less pyruvate and produces less ATP, and in some cases deploy ATP synthase to sustain the proton gradient at the expense of ATP to avoid a total collapse of membrane potential leading to apoptosis.

Relevance and importance of current work on the fundamental issue of the differential involvement of citrin and MCU-dependent matrix dehydrogenases in regulating basal metabolic setting of a cell is further supported by a recent preprint showing the involvement of aralar1 in boosting glycolysis and OxPhos upon stimulation in neurons[58]. Additionally, it was shown that cancer cells need a higher expression of citrin for proliferation and invasiveness, which can be counteracted by citrin down regulation[52]. In contrast to increased metabolism under citrin OE condition in these tumors, patients with citrin deficiency in form of type 2 citrullinemia struggle with various age-dependent metabolic pathologies that can be managed with proper diet[59]. Our present findings that point to the crucial role of MAS, as the main NADH shuttle in endothelial cells, in the regulation of mitochondrial bioenergetics are in line with previous reports in various tumor cells[52,60,61]. Under the basal condition, MAS continuously delivers NADH equivalents to the mitochondrial matrix, while maintaining glycolysis and pyruvate supply by

delivering $NAD^+$ into the cytosol[61]. Mitochondrial dehydrogenases oxidize TCA cycle substrates and fuel ETC by producing electron donors (NADH). The activity of both systems under resting conditions are controlled and synchronized by IMS and matrix $Ca^{2+}$ that are both supported by a basal $Ca^{2+}$ cycling in the MAMs (Fig. 8a–e, Fig. 9). Our data presented herein indicate that as soon as this basal subcellular $Ca^{2+}$ homeostasis/cycling is disturbed, the activity of the $Ca^{2+}$-regulated citrin is reduced, thus, delaying TCA cycle substrate supply and thus electron donor supply for the respiratory chain, while rerouting pyruvate to LDH. Our present data further indicate that under resting conditions, $Ca^{2+}$ influx into the mitochondrial matrix via the MCU is less important for regulation of basal mitochondrial bioenergetics than IMS $Ca^{2+}$ regulated citrin activity. This could serve as a preemptive mechanism under conditions of disturbed subcellular $Ca^{2+}$ homeostasis because the lack of citrin/MAS activity protects mitochondria against substrate overflow in light of reduced activity of $Ca^{2+}$ sensitive matrix dehydrogenases. One could argue that citrin is one order of magnitude more sensitive to $Ca^{2+}$[62] than matrix dehydrogenases and even small $Ca^{2+}$ fluctuations could then unnecessarily disturb mitochondrial function. However, our data indicate that matrix dehydrogenases do not sense small and short-term disturbances in basal subcellular $Ca^{2+}$ homeostasis and keep basal activity, thus putting citrin in charge of responding to slight and brief variations in basal $Ca^{2+}$ homeostasis. However, in order to achieve maximal mitochondrial ATP production, citrin and matrix dehydrogenases are synchronized upon stimulation by an $IP_3$-generating agonist (e.g., histamine) by the huge $Ca^{2+}$ transfer from the MAMs towards the IMS and mitochondrial matrix. In light of this cooperative action of citrin and the dehydrogenases, it appears beneficial for some cells to up-regulate citrin in addition to increasing MAMs and mitochondrial $Ca^{2+}$ uptake machinery in order to take full benefit of increased $Ca^{2+}$ supply by providing more fuel for the $Ca^{2+}$ sensitive dehydrogenases[52,63].

Our data point to a continuous PM-ER-MAM-IMS $Ca^{2+}$ flux to maintain mitochondrial surface/IMS $Ca^{2+}$ as a safeguard for the metabolic wiring of basal mitochondrial bioenergetics via

Ca$^{2+}$-controlled citrin. Disturbance of this subcellular Ca$^{2+}$ homeostasis instantly rewires mitochondrial bioenergetics into a pseudo-hypoxic state of enhanced glycolysis due to the lack of citrin activity while mitochondria struggle to preserve their $\Psi_{mito}$.

## Methods

**Cell culture and transfection.** EA.hy926 (provided by Dr. C.J.S. Edgell, University of North Carolina, Chapel Hill, NC, USA) and HeLa S3 (ATCC CCL-2.2) cells were grown in Dulbecco's Modified Eagle's Medium (DMEM) (Sigma-Aldrich; Vienna, Austria) containing 10% FCS, penicillin (100 U/ml), streptomycin (100 µg/ml), amphotericin (1.25 µg/ml), 1 g/L glucose and 4 mM glutamine in a humidified incubator (37 °C, 5% CO$_2$, 95% air). Origin of cells was confirmed by STR-profiling by the cell culture facility of ZMF (Graz). Cells were regularly tested for mycoplasma contamination and were negative. For all microscopy experiments with genetically encoded sensors or knockdown experiments using siRNA, cells were plated on 30 mm glass coverslips and transfected at 60-80% confluence (EA.hy926) or 40% confluence (HeLa) with 1 µg plasmid DNA encoding an appropriate sensor alone or together with siRNA using 2.5 µl (EA.hy926) or 3 µl (HeLa) of TransFast transfection reagent (Promega, Madison, WI, USA) in 1 ml of antibiotic-free medium (EA.hy926) or 1 ml serum- and antibiotic-free medium (HeLa) for 16–20 h. Afterward, the transfection media was replaced by full culture medium. All experiments were performed 40–48 h after transfection. Prior to experiments, cells were adjusted to room temperature and shortly kept in experimental storage buffer (2 mM Ca$^{2+}$, 138 mM NaCl, 1 mM MgCl$_2$, 5 mM KCl, 10 mM HEPES, 2.6 mM NaHCO$_3$, 0.44 mM KH$_2$PO$_4$, amino acid, and vitamin mix, 10 mM glucose, 2 mM L-glutamine, 1% Penicillin/Streptomycin, 1% Fungizone, pH adjusted to 7.4). Citrin siRNA (UAAAUAUGCACCUAGUUUUCCUtt), MCU siRNA (GCC AGAGACAGACAAUACUtt), IP$_3$R2 siRNA (GAGAAGGCUCGAUGCUGAG ACUUGAtt)[64], Orai1 siRNA (CGUGCACAAUCUCAACUCGtt), and scrambled siRNA control (UUCUCCGAACGUGUCACGU) were custom synthesized by Microsynth (Balgach, Switzerland).

**Quantitative PCR.** Total mRNA was isolated using PeqGOLD Total RNA Kit (VWR Peqlab, Leuven, Belgium), and reverse transcription was done using Applied Biosystems High Capacity cDNA Reverse Transcription kit (Thermo Fisher Scientific Baltics UAB, Vilnus, LT). qPCR was performed using Promega GOTaq® qPCR Master Mix (Madison, USA) on BIO-RAD CFX96™Real-Time System. Knockdown efficiency was determined using specific primers for citrin (forward: GCCCTTTAACTTGGCTGAGG; reverse: CCCAGACCAAACCTGTAGGC) and MCU (forward: CACTCGGGGCGGCTACTG; reverse: TGTACTACCGTCTCCC CTGG) and normalized to GAPDH.

**Western blot.** For KD validation, cells were seeded on 10 cm dishes, transfected with respective siRNAs and harvested 48 h post transfection. Rabbit polyclonal citrin antibody (Abcam, ab96303) at 1:1000 dilution and rabbit monoclonal MCU antibody (Cell Signaling Technology, D2Z3B, #14997) at 1:1000 dilution were used for immunoblotting. A 1:5000 dilution of goat anti-rabbit secondary antibody was used (Santa Cruz Biotechnology, sc-2054). For phosphorylated PDH assessment, cells on 10 cm dishes were incubated in experimental storage buffer for 20 min to adjust to room temperature, followed by 5 min incubation in 2 Ca$^{2+}$ buffer. After, cells were either incubated in 2 Ca$^{2+}$ buffer for 5 min, in 0 Ca$^{2+}$ for 5 min, or in 0 Ca$^{2+}$ buffer for 1 h. Following the incubation times, cells were washed with ice cold nominally Ca$^{2+}$ free buffer and harvested on ice. Phosphorylated PDH was blotted with 1:1000 dilution of rabbit mAb P-PDH S293 (Cell Signaling Technology, E4V9L, #37115), and total PDH with 1:1000 dilution of rabbit mAb PDH (Cell Signaling Technology, C54G1, #3205). A 1:5000 dilution of goat anti-rabbit secondary antibody was used (Santa Cruz Biotechnology, sc-2054). Broad Range (10-250 kDa) Color Prestained Protein Standard ladder (NEB, P7719S) was used in all blots.

**Live cell imaging.** All live-cell microscopy experiments were performed on an Olympus IX73 inverted microscope if not mentioned otherwise. The microscope is equipped with an UApoN340 40× oil immersion objective (Olympus, Japan) and a CCD Retiga R1 camera (Q-imaging, Canada). For illumination, a LedHUB® (Omnicron, Germany) equipped with 340, 385, 455, 470, and 550 nm LEDs in combination with CFP/YFP/RFP (CFP/YFP/mCherry-3X, Semrock, USA) or GFP (GFP-3035D, Semrock, USA) filter set was used. During the measurements cells were continuously perfused by a gravity-based perfusion system (NGFI, Graz, Austria). Data acquisition and control of the fluorescence microscope was performed using Visiview 4.2.01 (Visitron, Germany).

**Ca$^{2+}$ measurements.** EA.hy926 cells were perfused with calcium containing physiological buffer (2 mM Ca$^{2+}$, 135 mM NaCl, 1 mM MgCl$_2$, 5 mM KCl, 10 mM HEPES, 10 mM glucose, pH adjusted to 7.4) and the basal calcium level were recorded for 2–3 min. Then, the buffer was changed to Ca$^{2+}$ free buffer (138 mM NaCl, 1 mM MgCl$_2$, 5 mM KCl, 10 mM HEPES, 0.1 mM EGTA, 10 mM glucose, pH adjusted to 7.4) in perfusion and the corresponding changes were recorded in cytosol, ER, IMS and mitochondria using FURA2-AM, D1ER[65], IMS-GEM-

GECO1[66,67], and 4mtD3cpv[68] Ca$^{2+}$ indicators respectively. For FURA2-AM loading, cells were incubated in storage buffer with 3 µM FURA2-AM for 30 min at room temperature and washed with fresh storage buffer. FURA2-AM was sequentially excited with 340 nm and 385 nm LEDs and emission collected with GFP emission filter set. 4mtD3cpv was excited with 455 nm LED and emission simultaneously collected with CFP/YFP/RFP filter set and 505dcxr beam-splitter (Semrock, USA). The same set up as for 4mtD3cpv was used for imaging cells expressing D1ER. IMS-GEM-GECO1 was excited with 385 nm LED and emission collected at 480 nm and 530 nm using a CFP/YFP/RFP filter set and 505dcxr beam-splitter.

Basal cytosolic Ca$^{2+}$ level, as well as the corresponding drop due to extracellular Ca$^{2+}$ removal, were normalized using the minimal Ca$^{2+}$ level achieved by Ca$^{2+}$ free buffer with 1 mM EGTA and 4 µM Ionomycin, a Ca$^{2+}$ specific ionophore. Basal ER Ca$^{2+}$ level, as well as the corresponding drop due to extracellular Ca$^{2+}$ removal, were normalized using maximal releasable ER Ca$^{2+}$ achieved with 100 µM histamine, an IP$_3$ generating agonist, and 15 µM BHQ, a SERCA inhibitor, in Ca$^{2+}$ free buffer. IMS and mitochondrial Ca$^{2+}$ concentrations were determined via achieving minimum and maximum Ca$^{2+}$ signals by varying buffer solutions in the presence of ionomycin as previously described[68]. Shortly, after recording basal Ca$^{2+}$ level for 2 min, cells were perfused with 0 Ca$^{2+}$ buffer containing 4 µM ionomycin for 15 min to achieve a minimal Ca$^{2+}$ concentration. After, cells were perfused with 2 Ca$^{2+}$ in the presence of ionomycin to achieve the maximal. For some experiments, the initial ionomycin addition resulted in higher peak, in these cases, the highest ratio was taken as maximum. For the mitochondrial Ca$^{2+}$ uptake experiment, cells were perfused with 2 Ca$^{2+}$ buffer for 1.5 min and stimulated with 100 µM histamine for 2 min. The maximum ratio change achieved was quantified.

**ER-plasma membrane and ER-mitochondria contact sites.** ER-plasma membrane contact cites were imaged using SIM setup composed of a 405-, 488-, 515-, 532- and a 561-nm excitation laser introduced at the back focal plane inside the SIM box with a multimodal optical fiber. For super-resolution, a CFI SR Apochromat TIRF ×100-oil (NA 1.49) objective was mounted on a Nikon-Structured Illumination Microscopy (N-SIM) System with standard widefield and SIM filter sets and equipped with two Andor iXon3 EMCCD cameras mounted to a two camera imaging adapter (Nikon Austria, Vienna, Austria). For calibration and reconstruction of SIM images the Nikon software Nis-Elements (Nikon Austria, Vienna, version 4.51.00 64 bit) was used. Prior to each measurement, laser adjustment was checked by projecting the laser beam through the objective at the top cover of the bright field arm of the microscope.

Total internal reflection microscopy was done using the N-SIM TIRF grating for 488 nm laser wavelength. Cells expressing an inactive version of ERAT4.01[69] and grown on 1.5H high-precision glass coverslips were incubated in 0 Ca$^{2+}$ or 2 Ca$^{2+}$ buffer for 5 min, placed with the respective buffer into the live-cell chamber, and imaged. Reconstruction of SIM images was done using the NIS-Elements software. For further analysis, the images were background subtracted using the Mosaic suite background subtractor, median filtered with a radius of 3 pixel, and auto thresholded using an Otsu threshold for segmentation. The segmented images were analyzed using Fiji with the included particle analyzer to measure size and count of ER patches. The total area of the cells was analyzed manually by measuring the cell size with an ROI.

For ER-mitochondria contacts, EA.hy926 cells were transfected with mitochondrial matrix targeted DsRed fluorescent protein and on the next day infected with a virus carrying an ER targeted inactive version of CFP-YFP FRET based ATP sensor ERAT4.01. Cells were imaged on a confocal spinning disk microscope (Axio Observer.Z1 from Zeiss, Gottingen, Germany) equipped with a 100x objective lens (Plan-Fluor x100/1.45 Oil, Zeiss), a motorized filter wheel (CSUX1FW, Yokogawa Electric Corporation, Tokyo, Japan) on the emission side, AOTF-based laser merge module for laser line 405, 445, 473, 488, 561, and 561 nm (Visitron Systems) and a Nipkow-based confocal scanning unit (CSU-X1, Yokogawa Electric Corporation). The ERAT1.03 and mitochondrial DsRed were alternately excited with 488 and 561 nm laser lines, respectively, and emissions were acquired at 530 and 600 nm using a charged CCD camera (CoolSNAP-HQ, Photometrics, Tucson, AZ, USA). Z-stacks of both channels in 0.2 µm increments were recorded. VisiView acquisition software (Universal Imaging, Visitron Systems) was used to acquire the imaging data. Images were blind deconvoluted with NIS-elements v5.20.02 (Nikon, Austria). The colocalization was determined on a single-cell level using ImageJ and the plugin coloc2. The Pearson coefficient and the Costes tresholded Manders 1 or 2 coefficients were calculated.

**Mitochondrial ATP and membrane potential measurements.** Mitochondrial ATP in EA.hy926 and HeLa cells was measured using mitochondrial matrix targeted AT1.03[30] (gift from Hiromi Imamura, Kyoto University, Kyodai Graduate School of Biostudies, Japan), a genetically encoded ATP sensor. The sensor was excited with 455 nm LED and emission collected at 480 nm and 530 nm using a CFP/YFP/RFP filter set and 505dcxr beam-splitter. Cells were perfused with Ca$^{2+}$ buffer alone or Ca$^{2+}$ buffer followed by 0 Ca$^{2+}$ buffer. Mitochondrial ATP production or consumption was assessed by the addition of 2 µM oligomycin (Sigma-Aldrich, Vienna, Austria) in perfusion. Decrease of the ATP below basal level after oligomycin addition was considered to be ATP production, whereas increase of the ATP above the basal level was considered to be ATP consumption by the ATP

synthase. For the rescue of ATP production experiments, cell were sequentially perfused with $Ca^{2+}$ buffer (2 min), 0 $Ca^{2+}$ buffer (6 min), nutrient supplemented $Ca^{2+}$ buffer (experimental storage buffer, 10 min), and $Ca^{2+}$ buffer (3 min) before perfusing with 2 µM oligomycin. For simultaneous membrane potential measurements, cells transfected with mtAT1.03 sensor were incubated with 20 nM TMRM for at least 30 min in experimental storage buffer at room temperature right before measurements. All of the buffers used afterward in perfusion contained the same concentration of TMRM. TMRM data was used qualitatively to determine the direction of ATP synthase after oligomycin addition and to correlate with ATP measurements. For mitochondrial ATP production after histamine addition—cells were perfused in $Ca^{2+}$ buffer for 2 min, after which cells were perfused with 10 µM histamine for 6–7 min. Change in mitochondrial ATP 6 min after histamine addition was quantified.

For mitochondrial ATP measurements with BAPTA-AM and EGTA-AM, EA.hy926 cells were incubated with 50 µM solution of respective chelators in experimental storage buffer for 30 min, washed with fresh buffer, and measured afterward on Olympus IX73 inverted microscope while perfused with $Ca^{2+}$ containing buffer.

For mitochondrial ATP measurements with Xestospongin C (10 µM)[45], Pyrozole 6 (3 µM), Pyrozole 10 (3 µM)[46] and combination of Pyrozole 6 and 10 (3 µM each), EA.hy926 cells were incubated with respective inhibitors or DMSO in 2 $Ca^{2+}$ buffer for 15 min, washed and imaged on an iMic inverted and advanced fluorescent microscope using a x20 magnification objective (Fluar x20/0.75, Zeiss, Göttingen, Germany) with a motorized sample stage (TILL Photonics, Graefling, Germany) without perfusion. Concentrations used should represent 75% inhibition or higher and were taken from respective publications[45,46]. After baseline recording, 2X oligomycin was added for a final concentration of 2 µM. For control and acquisition, Live Acquisition 2 (TILL Photonics) software was used. The sensor was excited at wavelength of 430 nm; emission was collected simultaneously at 535 and 480 nm using an optical beam-splitter (Dichroic 69008ET-ECFP/EYFP/mCherry). Data processing was performed with the Offline Analysis application (TILL Photonics). The inhibitor concentrations were corresponding to 60–70% inhibition from respective source publications.

**Mitochondrial NADH autofluorescence measurements**. Mitochondrial NADH autofluorescence was monitored using 340 nm LED as previously described[70]. Shortly, EA.hy926 cells were perfused with 2 $Ca^{2+}$ buffer, followed by perfusion with 1 µM FCCP until the signal flattens, and then perfused with 2 µM rotenone until a plateau is reached. Mitochondrial NADH production was quantified as maximum NADH autofluorescence achieved by Complex I inhibition with 2 µM rotenone, normalized to the basal autofluorescence after subtracting the minimal autofluorescence reached by 1 µM FCCP. For $Ca^{2+}$ free experiments, cells were perfused with 0 $Ca^{2+}$ buffer after 2 min in 2 $Ca^{2+}$ buffer, with subsequent additions of FCCP and rotenone in 0 $Ca^{2+}$ buffer. For the rescue experiments, cells were switched back to 2 $Ca^{2+}$ buffer for 10 min after 0 $Ca^{2+}$ buffer.

**Measurement of mitochondrial respiration**. 35,000 cells were plated on XF96 polystyrene cell culture microplates (Seahorse®, Agilent, CA, USA) 24 h before the experiment. For the KD experiments, cells were transfected in 10 cm dishes overnight and seeded on XF96 polystyrene cell culture microplates 24 h before the experiment. Thirty minutes before the measurement, cell medium was changed to XF assay medium supplemented with 1 mM sodium pyruvate, 2 mM glutamine and 5.5 mM D-glucose and incubated in non-$CO_2$ 37 °C incubator. Prior to loading the plate, XF assay media was refreshed and $Ca^{2+}$ free assay media was added to respective wells. XF96 extracellular flux analyzer was used to measure oxygen consumption rate (OCR) and extracellular acidification rate (ECAR). OCR (pmol O2/min) and ECAR (mpH/min) values were normalized to protein content. For the rescue, ECAR was assessed following injection of 25 µl 7x $Ca^{2+}$ buffer after basal ECAR in 0 $Ca^{2+}$ was measured. Protein concentration of each well was determined with the Pierce™ BCA Protein Assay Kit (Thermo Scientific, Rockford, IL, USA).

**Live cell metabolite measurements**. All live cell metabolite measurements were done in EA.hy926 cells. Mitochondrial citrate and pyruvate were measured using genetically-encoded sensors mito-Citron1[35] (gift from Robert Campbell, Addgene plasmid # 134305) and mito-PyronicSF[36] (gift from Luis Felipe Barros, Addgene plasmid # 124813) respectively. Both sensors were excited with 470 nm LED and emission collected with a GFP filter set. After acquiring baseline readout in 2 $Ca^{2+}$ or 0 $Ca^{2+}$ buffers, cells were switched to 1 mM pyruvate in 2 $Ca^{2+}$ or 0 $Ca^{2+}$ buffer in perfusion. For experiments in 0 $Ca^{2+}$ condition, cells were kept 3 min in 0 $Ca^{2+}$ buffer prior to the start of the measurement to bring total time in 0 $Ca^{2+}$ to 5 min. For mitochondrial pyruvate supply experiments, cells were perfused in 0 glucose + /− $Ca^{2+}$ buffer for 5 min before the start of the measurement. After, 2 min baseline was taken in 0 glucose buffer with subsequent switch to ten glucose-containing buffer. The plateau phase in the respective ten glucose buffer was quantified.

Cytosolic lactate was measured with Laconic[71] (gift from Luis Felipe Barros, Addgene plasmid # 44238). Cells expressing Laconic were excited with 455 nm LED and emission collected at 480 nm and 530 nm using a CFP/YFP/RFP filter set

and 505dcxr beam-splitter. After acquiring basal readout in 2 $Ca^{2+}$, the buffer was changed to 0 $Ca^{2+}$ in perfusion and measured for 5–6 min. The increase in lactate after 5 min was quantified. For the rescue experiment, after 0 $Ca^{2+}$ buffer, cells were perfused with 2 $Ca^{2+}$ buffer for 10 min.

Cytosolic glucose was measured using FLII12Pglu-700µδ6[72] (gift from Wolf Frommer, Addgene plasmid # 17866). Cells expressing FLII12Pglu-700µδ6 were excited with 455 nm LED and emission collected at 480 nm and 530 nm using a CFP/YFP/RFP filter set and 505dcxr beam-splitter. Cells were perfused with either 2 $Ca^{2+}$ or 0 $Ca^{2+}$ buffers, after baseline recording, the perfusion buffer was changed to respective glucose-free buffer, after the signal reached a new flat baseline, glucose was re-added in perfusion.

Cytosolic pyruvate was measured with Pyronic[43] (gift from Luis Felipe Barros, Addgene plasmid # 51308). Cells expressing Pyronic were excited with 455 nm LED and emission collected at 480 nm and 530 nm using a CFP/YFP/RFP filter set and 505dcxr beam-splitter. After acquiring a baseline readout in 2 $Ca^{2+}$ or 0 $Ca^{2+}$ buffers, Lactate dehydrogenase (LDH) inhibitor GSK 2837808A[73] and Monocarboxylate transporter (MCT) inhibitor AR-C155858[74] were added in perfusion. Pyruvate accumulation was quantified after 5 min application.

**Cytosolic NADH/NAD$^+$ ratio**. Cytosolic NADH/NAD ratio was measured using Peredox[38] (gift from Gary Yellen, Addgene plasmid # 32383). EA.hy926 cells expressing Peredox were excited with 385 nm and 550 nm LEDs and emission collected using a CFP/YFP/RFP filter set and 565 LPXR beam-splitter (Semrock, USA). Basal readout in 2 $Ca^{2+}$ buffer, as well as the change after perfusion with 0 $Ca^{2+}$ buffer, was recorded. For rescue experiments, cells were further perfused for 10 min with 2 $Ca^{2+}$ buffer. After every measurement, the green to red emission ratio, representing NADH/NAD$^+$ ratio, was normalized to maximum (1.0) and minimum (0) using 10 mM lactate and 10 mM pyruvate, respectively.

**Cell viability and apoptosis assays**. Cells were seeded on 96-well plates and left in an incubator under standard conditions for 24 h to settle. Afterward, cell viability was assessed with the CellTiter-Blue® Cell Viability Assay (Promega, Madison, USA) and the activity of caspase 3/7 was investigated as an indicator for apoptosis with the Caspase-Glo® 3/7 Assay (Promega, Madison, USA). Both assays were performed as per company instructions. Staurosporine was applied in a concentration of 1 µM for 16 h as a positive control for reduced cell viability and increased apoptosis. Finally, cells were washed and the protein concentration of each well was determined with the Pierce™ BCA Protein Assay Kit (Thermo Scientific, Rockford, IL, USA) in order to normalize the results.

**Statistical analysis and reproducibility**. Number of independent experiments is indicated in each figure legend along with the used statistical test and $p$-value. Statistical analyses, including student's $t$-test and analysis of variance (ANOVA) with Tukey post hoc test, were performed on GraphPad Prism software version 5.04 (GraphPad Software, San Diego, CA, USA) or Microsoft Excel (Microsoft Office 2013). All box and whisker plots show minimum to maximum values with the central line showing the median and boxes extending from 25–75% of the data set. All scatter plots show single cells. Differences with $p < 0.05$ were considered to be statistically significant.

**Reporting summary**. Further information on research design is available in the Nature Research Reporting Summary linked to this article.

## Data availability

The source data used to generate the graphs and charts in the main manuscript and in the Supplementary Figures are available in Supplementary Data 1. Any additional data and resources used in the current manuscript are available from the corresponding author on reasonable request. Genetically encoded sensors used in the current manuscript that are available from Addgene: mito-Citron1[35] (gift from Robert Campbell, Addgene plasmid # 134305), mito-PyronicSF[36] (gift from Luis Felipe Barros, Addgene plasmid # 124813), Laconic[71] (gift from Luis Felipe Barros, Addgene plasmid # 44238), FLII12Pglu-700µδ6[72] (gift from Wolf Frommer, Addgene plasmid # 17866), Pyronic[43] (gift from Luis Felipe Barros, Addgene plasmid # 51308), Peredox[38] (gift from Gary Yellen, Addgene plasmid # 32383), 4mtD3cpv[68] (gift from Amy Palmer and Roger Tsien, Addgene plasmid #36324). Constructs not listed above are available upon reasonable request from the corresponding author.

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

## Acknowledgements

The authors wish to thank Mrs. Anna Schreilechner, BSc, for her great work in maintaining cell culture. We are greatly indebted to Dr. Edgell (University of North Carolina, Chapel Hill, NC, U.S.A.) for providing us with the human endothelial cell line EA.hy926. This work was supported/funded by the Austrian Science Fund (FWF) (DK-MCD W1226), MEFO Graz, BioTechMed Graz, Nikon Austria, and the Medical University of Graz. Z.K. and F.E.O. are supported by the FWF (DK-MCD W1226), and M.H. by MEFO Graz in course of the doctoral college DK-MCD.

## Author contributions

Conceptualization and design of the work, Z.K. and W.F.G.; acquisition of data, Z.K., F.E.O., M.H., B.G., R.R.; analysis of data, Z.K., B.G.; interpretation of data, Z.K., W.F.G.; writing—original draft preparation, Z.K.; writing–review and editing, B.G., R.M., W.F.G.; All authors have read and agreed to the published version of the manuscript.

## Competing interests

The authors declare no competing interests.
