## [Transparent Peer Review File · Communications Biology]

Reviewers' comments:

Reviewer #1 (Remarks to the Author):

In this study, Koshenov et al. report an interesting role of Citrin/SLC25A13 in regulating metabolic rewiring in response to changes in cellular Ca²⁺ homeostasis. The authors induced dysregulation of basal subcellular Ca²⁺ homeostasis by removing Ca²⁺ from the extracellular milieu. Using a variety of genetically encoded probes, Koshenov and colleagues demonstrate that small changes in the MAMs and IMS Ca²⁺ levels can induce metabolic rewiring by stimulating a state of "pseudo hypoxia" which in turn leads to higher glycolysis and lactate accumulation. Further, the authors show that citrin but not MCU plays an important role in connecting perturbations in Ca²⁺ homeostasis to metabolic rewiring under the experimental conditions tested by them. Overall, it is a well-conceived study wherein most of the conclusions are supported by well controlled experiments. However, there are some concerns that should be addressed for strengthening few interpretations and improving the quality of the study. Refer to the below points for details:

1. For evaluating the role of MAMs and SOCE Ca²⁺ flux in regulating mitochondrial bioenergetics, the authors have drawn conclusions on the basis of pharmacological inhibitors only. These inhibitors may have non-specific effects especially when used at higher concentrations. This data is one of the significant observations of the study. Therefore, some molecular validation is required for establishing the role of MAMs and SOCE Ca²⁺ flux. It can be easily performed by silencing IP3R2 and Orai1 as they are key MAMs and SOCE Ca²⁺ flux players.
2. Can pyruvate supplementation enhance NADH production in no extracellular Ca²⁺ condition and bring it to the levels observed in 2mM extracellular Ca²⁺ condition (not supplemented with pyruvate)? This experiment is critical for interpreting that mitochondrial dehydrogenase activity is unaffected by changes in extracellular Ca²⁺ levels.
3. What happens to mitochondrial pyruvate levels upon citrin knockdown? Although removal of extracellular Ca²⁺ does not affect mitochondrial pyruvate levels (Figure 3f), it should be analyzed upon citrin silencing as well.
4. On page no. 15, authors write "supporting the hypothesis that reduced pyruvate availability for mitochondria under reduced IMS Ca²⁺ is one of the reasons for worsened mitochondrial bioenergetics when subcellular/IMS Ca²⁺ homeostasis is disturbed". But the data presented in Figure 3f suggests that mitochondrial pyruvate levels are not altered during extracellular Ca²⁺ deprivation. If that is the case then on what basis authors are postulating that reduced IMS Ca²⁺ (which is happening downstream of extracellular Ca²⁺ deprivation) would alter mitochondrial bioenergetics by reducing mitochondrial pyruvate levels. Please clarify this.
5. Why and how mitochondrial membrane potential is decreased in the cells with perturbed basal subcellular Ca²⁺ homeostasis? It should be elaborated and discussed in light of literature and/or data presented in this study.
6. Some of the references are not complete. For example: the complete details of the reference # 40 are not included in the reference list. Please recheck the referencing section.

Reviewer #2 (Remarks to the Author):

In this manuscript, Koshenov and colleagues show that perturbation of basal intracellular Ca²⁺ is associated with mitochondrial bioenergetic defects and metabolic rewiring. In this context, the authors identify citrin as a major player. Overall, the hypothesis proposed here is new and provocative.

However, some weaknesses prevent the publication in the present form, and must be thoroughly considered:

- Ca²⁺ supplementation rescues mitochondrial ATP after oligomycin addition (Figure 2c). The Authors should verify whether mitochondrial and cytosolic NADH production, cytosolic lactate and basal ECAR are rescued, too.
- Figure 3d-e: The phosphorylation levels of PDH should be assessed to corroborate the conclusion that PDH activity is not involved in the control of mitochondrial bioenergetics by perturbation of basal Ca²⁺. Furthermore, the authors should take into account that PDH activity is indirectly regulated not only by Ca²⁺ but also by ATP and substrates, the concentrations of which are altered in 0 Ca²⁺ condition. Please, discuss this point.
- In figure 5i, it is not clear why MCU KD triggers a significant difference in mitochondrial ATP level after oligomycin addition in the presence compare to the absence of external Ca²⁺ (Figure 5i). Please check and show MCU protein expression upon MCU KD.
- The Authors find that LDH and MCT use more pyruvate during IMS Ca²⁺ imbalance or in the absence of citrin, thus diminishing pyruvate availability for mitochondria. However, when they measured basal cytosolic pyruvate they did not find any difference between Ca²⁺-treated and untreated cells (Figure 6a). They refer this effect to the lack of sensitivity of their probe that required them to determine pyruvate accumulation after MCT and LDH inhibition (Figure 6b). However, to prove that citrin controls basal pyruvate availability the authors should measure basal lactate in control and citrin KD cells in the presence and in the absence of external Ca²⁺.
- Does citrin KD affect mitochondrial Ca²⁺ uptake? Does MCU overexpression rescue the metabolic defect of citrin KD? Please perform experiments accordingly.
- The fact that perturbation of subcellular Ca²⁺ homeostasis does not affect the activity of the mitochondrial pyruvate dehydrogenase is not sufficient to conclude that Ca²⁺ perturbation does not affect mitochondrial dehydrogenases. Please modify this assumption throughout the manuscript.

Reviewer #3 (Remarks to the Author):

The work of Koshenov and colleagues has been designed to investigate principal regulating mechanisms of basal mitochondrial bioenergetics as fundamental processes for conditioning of cellular metabolic wiring. They found that Ca²⁺ fluxes from the ER-mitochondria contact sites activate the mitochondrial metabolism and keep mitochondria energized. Furthermore, they identified citrin as the primary regulator of this Ca²⁺-dependent regulation. Finally, they affirm that manipulation of these Ca²⁺ dynamics may be useful to reprogram the mitochondrial metabolism in resting and pathological condition.

At today, it is widely accepted that intracellular Ca²⁺ variations, in particular modification of the Ca²⁺ transmission between ER and mitochondria, modulate the mitochondrial energy production with great impact with the overall cellular metabolism. In the first figures, authors focus their attention to this aspect. They give another demonstration of this, they establish a protocol that minimally perturbs basal subcellular Ca²⁺ homeostasis affecting the mitochondria and sub-mitochondrial compartments and reveal important aspects of both intracellular and extracellular Ca²⁺ in preserving the mitochondrial and cellular energetics.

However, in my opinion, I think that the primary finding of this work is to have identified citrin as the responsible for the metabolic rewiring during disturbed subcellular Ca²⁺ homeostasis. At demonstration the title of the manuscript is: "Citrin mediated metabolic rewiring in response to altered basal subcellular Ca²⁺ homeostasis".

Thus, I would encourage authors to perform new experiments aimed to demonstrate these findings. In particular authors should deep investigate about the importance of citrin in Ca²⁺ signaling. No experiments have been performed to demonstrate that citrin regulates Ca²⁺ at different compartments (cytosol, ER and mitochondria) as well as they lack to demonstrate that citrin alter the

Ca²⁺ transmission between ER and mitochondria. I think these experiments are crucial since authors focus their attention to Ca²⁺ and observe that citrin deletion has similar effects like those observed in the condition with MCU deleted.

Furthermore, authors should include more evidences of the alteration of the mitochondrial energetic in the citrin and MCU KD conditions by performing experiments like those reported in Figure 2.

Authors identify Citrin a responsible for cytosolic and mitochondrial metabolic rewiring. I think they should also explore this possibility in pathologic context. Does exist a particular disease in which citrin is downregulated or over-expressed? Is it possible to counteract the disease phenotype by modulating the citrin expression?

Finally, in figure 6 authors demonstrate that citrin controls pyruvate availability for mitochondria. In the subsequent figure authors demonstrate that mitochondrial bioenergetics are influenced by basal MAM Ca²⁺ flux. Here, no experiments have been conducted with citrin. These findings seem to be unrelated to each other.

Minor points:

- Please simplify the schematic illustration of Figure 8.
- transient knockdown of citrin should be also demonstrated by immunoblot
- Citrin KD profoundly reduces the energetic production. Authors should verify if this condition also affects the cell growth and cell death processes like apoptosis.

Response to the referee's comments:

We thank the editor and the referees very much for the insightful comments and the fair and valuable suggestions that helped us very much in improving our work. We apologize for the rather long time needed for this revision, but our non-European coworkers took the relaxation phase of the COVID situation to visit their home countries and families whose they could not see for a long time. We have addressed each single point raised with very carefully and have performed respective experiments if possible.

Please find below our point-to-point response to the individual points raised:

Reviewers' comments:

Reviewer #1 (Remarks to the Author):

In this study, Koshenov et al. report an interesting role of Citrin/SLC25A13 in regulating metabolic rewiring in response to changes in cellular Ca²⁺ homeostasis. The authors induced dysregulation of basal subcellular Ca²⁺ homeostasis by removing Ca²⁺ from the extracellular milieu. Using a variety of genetically encoded probes, Koshenov and colleagues demonstrate that small changes in the MAMs and IMS Ca²⁺ levels can induce metabolic rewiring by stimulating a state of "pseudo hypoxia" which in turn leads to higher glycolysis and lactate accumulation. Further, the authors show that citrin but not MCU plays an important role in connecting perturbations in Ca²⁺ homeostasis to metabolic rewiring under the experimental conditions tested by them. Overall, it is a well-conceived study wherein most of the conclusions are supported by well controlled experiments.

We thank the reviewer for the supportive and kind words and have addressed the points raised as follows:.

However, there are some concerns that should be addressed for strengthening few interpretations and improving the quality of the study. Refer to the below points for details:

1. For evaluating the role of MAMs and SOCE Ca²⁺ flux in regulating mitochondrial bioenergetics, the authors have drawn conclusions on the basis of pharmacological inhibitors only. These inhibitors may have non-specific effects especially when used at higher concentrations. This data is one of the significant observations of the study. Therefore, some molecular validation is required for establishing the role of MAMs and SOCE Ca²⁺ flux. It can be easily performed by silencing IP3R2 and Orai1 as they are key MAMs and SOCE Ca²⁺ flux players.

We would like to thank the reviewer for this valuable suggestion that further challenges our previous results by an alternative approach. Accordingly, we have performed additional mitochondrial ATP measurements in EA.hy926 cells with siRNA (**lines 553-554**) depleted IP₃R2 and Orai1. The results support our previous assumption and have been added as part of **Fig. 8 d-e, and described these new findings in lines 390-396.**

2. Can pyruvate supplementation enhance NADH production in no extracellular Ca²⁺ condition and bring it to the levels observed in 2mM extracellular Ca²⁺ condition (not supplemented with pyruvate)? This experiment is critical for interpreting that mitochondrial dehydrogenase activity is unaffected by changes in extracellular Ca²⁺ levels.

Thank you for pointing to this important aspect of the manuscript. We have compared NADH production in 2Ca vs 0Ca with pyruvate and separately 2Ca vs 0Ca without pyruvate, which showed that absence of extracellular Ca^{2+} doesn't impact mitochondrial NADH production when the cells have pyruvate available, and only affect it when the cells have only glucose available. It is difficult to compare cells supplemented with pyruvate to the ones that are not because pyruvate supplementation increases basal NADH autofluorescence (Figure 3a), making it problematic to normalize and to compare. We hope for the referee's understanding on this point.

3. What happens to mitochondrial pyruvate levels upon citrin knockdown? Although removal of extracellular Ca^{2+} does not affect mitochondrial pyruvate levels (Figure 3f), it should be analyzed upon citrin silencing as well.

We thank the referee for this valuable suggestion. As in figure 3f we have used an intensimetric sensor we can't judge the basal pyruvate levels. To address the referee's point, we have measured mitochondrial pyruvate levels when we went from 0 to 10 mM glucose buffer in control, citrin KD, and MCU KD cells under 2 Ca^{2+} and 0 Ca^{2+} conditions, to visualize mitochondrial pyruvate supply. These experiments have shown that mitochondrial pyruvate supply is controlled by citrin activity as well as IMS Ca^{2+} . These results are shown in **Fig. 7 c-e, described in lines 332-341 and discussed them in lines 454 to 455**, experimental procedure described in materials and methods section, **lines 712-715**.

4. On page no. 15, authors write "supporting the hypothesis that reduced pyruvate availability for mitochondria under reduced IMS Ca^{2+} is one of the reasons for worsened mitochondrial bioenergetics when subcellular/IMS Ca^{2+} homeostasis is disturbed". But the data presented in Figure 3f suggests that mitochondrial pyruvate levels are not altered during extracellular Ca^{2+} deprivation. If that is the case then on what basis authors are postulating that reduced IMS Ca^{2+} (which is happening downstream of extracellular Ca^{2+} deprivation) would alter mitochondrial bioenergetics by reducing mitochondrial pyruvate levels. Please clarify this.

Thank you for the awareness and for this important correction. Our initial reasoning was based on results showing that LDH and MCT receive more pyruvate in citrin KD and 0Ca, and on the fact that there is more lactate accumulation in 0Ca and citrin KD. Therefore, we postulated that mitochondria under these conditions would receive less pyruvate as it is redirected towards lactate production as a result of reduced citrin levels (KD) or activity (0Ca). Additionally, we have shown that pyruvate supplementation rescues the mitochondrial ATP production, which indirectly proves the need for pyruvate in 0Ca. Based on referee's current and previous comment, we directly measured mitochondrial pyruvate levels when we go from 0 glucose buffer to 10 mM glucose in the presence and absence of extracellular Ca^{2+} in control, citrin KD, and MCU KD cells. This experiment confirmed that indeed the glycolytic flux from glucose to mitochondrial pyruvate is under control of citrin activity and can be modulated by IMS/extracellular Ca^{2+} . These results are now demonstrated in **Fig. 7 c-e, described in lines 332-341 and discussed them in lines 454 to 460**.

5. Why and how mitochondrial membrane potential is decreased in the cells with perturbed basal subcellular Ca^{2+} homeostasis? It should be elaborated and discussed in light of literature and/or data presented in this study.

We thank the referee for this comment. We have addressed this issue already in the previous version (lines 129 to 136) and now have added new aspects at **lines 457-460**.

6. Some of the references are not complete. For example: the complete details of the reference # 40 are not included in the reference list. Please recheck the referencing section.

We apologize for the mistakes and have rechecked and corrected the referencing section.

Reviewer #2 (Remarks to the Author):

In this manuscript, Koshenov and colleagues show that perturbation of basal intracellular Ca²⁺ is associated with mitochondrial bioenergetic defects and metabolic rewiring. In this context, the authors identify citrin as a major player. Overall, the hypothesis proposed here is new and provocative. However, some weaknesses prevent the publication in the present form, and must be thoroughly considered:

We are thankful for the referee on the fair and kind comments and have addressed each individual point as indicated below:

- *Ca²⁺ supplementation rescues mitochondrial ATP after oligomycin addition (Figure 2c). The Authors should verify whether mitochondrial and cytosolic NADH production, cytosolic lactate and basal ECAR are rescued, too.*

We thank the referee for this valuable comment. We have performed the rescue experiments suggested by the reviewer and can confirm that Ca²⁺ supplementation/re-addition can rescue reduced mitochondrial NADH production, increased cytosolic NADH/NAD⁺ ratio, increased cytosolic lactate levels as well as increased basal ECAR. These new results were added in **Supplementary Fig. 2 a-b, Supplementary Fig. 4 a-c**, and described in **lines 136-138 and 218-219**. Procedures are described in materials and methods section, **lines 689-690, 700-703, 720-721, 739**

- *Figure 3d-e: The phosphorylation levels of PDH should be assessed to corroborate the conclusion that PDH activity is not involved in the control of mitochondrial bioenergetics by perturbation of basal Ca²⁺. Furthermore, the authors should take into account that PDH activity is indirectly regulated not only by Ca²⁺ but also by ATP and substrates, the concentrations of which are altered in 0 Ca²⁺ condition. Please, discuss this point.*

This is an important point. Accordingly, the phosphorylation status of PDH was assessed and included in **figure 3 h-i** as well as **supplementary figure 3**. The regulation by substrate and ATP was mentioned in results section, **lines 178-183** and discussed it in **lines 431 to 435**. The methods described in **lines 564-574**.

- *In figure 5i, it is not clear why MCU KD triggers a significant difference in mitochondrial ATP level after oligomycin addition in the presence compare to the absence of external Ca²⁺ (Figure 5i). Please check and show MCU protein expression upon MCU KD.*

Thank you for this insightful suggestion. As MCU facilitates Ca²⁺ uptake into the matrix, MCU KD would only affect the matrix residing dehydrogenases and not IMS localized citrin, thus going from 2Ca²⁺ to 0Ca²⁺ buffer in MCU KD would additionally decrease IMS Ca²⁺ levels as we have shown and

add the effect of reduced MAS activity to reduced matrix dehydrogenase activity achieved by MCU KD. Additionally, mitochondrial pyruvate measurements showed that while MCU KD doesn't affect mitochondrial pyruvate supply in the presence of extracellular calcium, removal of extracellular calcium does (**Fig. 7 c-e** and described it in **lines 332-341**). We have added western blot data for MCU KD in **supplementary figure 5**, methods described in **lines 564-574**.

• The Authors find that LDH and MCT use more pyruvate during IMS Ca²⁺ imbalance or in the absence of citrin, thus diminishing pyruvate availability for mitochondria. However, when they measured basal cytosolic pyruvate they did not find any difference between Ca²⁺-treated and untreated cells (Figure 6a). They refer this effect to the lack of sensitivity of their probe that required them to determine pyruvate accumulation after MCT and LDH inhibition (Figure 6b). However, to prove that citrin controls basal pyruvate availability the authors should measure basal lactate in control and citrin KD cells in the presence and in the absence of external Ca²⁺.

We thank the referee for this valuable comment. We have addressed a similar point for reviewer 1 and measured mitochondrial pyruvate directly when going from 0 glucose to 10 mM glucose buffer to assess the pyruvate that reaches mitochondria under control, citrin KD, and MCU KD conditions under 2Ca²⁺ and 0Ca²⁺ **Fig. 7 c-e** and described it in **lines 332-341**. This experiment confirmed that indeed the glycolytic flux from glucose to mitochondrial pyruvate is under control of citrin activity and can be modulated by IMS/extracellular Ca²⁺, while MCU/matrix dehydrogenases do not seem to influence it under this experimental protocol.

• Does citrin KD affect mitochondrial Ca²⁺ uptake? Does MCU overexpression rescue the metabolic defect of citrin KD? Please perform experiments accordingly.

We thank the referee for this suggestion. We have compared mitochondrial Ca²⁺ uptake under stimulation with IP₃ generating agonist in control vs citrin KD cells and confirmed that citrin KD failed to affect mitochondrial Ca²⁺ uptake. These results are shown in **supplementary figure 9**

We have further tested whether or not MCU overexpression can rescue metabolic defects of citrin KD by repeating 2 main experiments with MCU OE on mitochondrial ATP production, shown in **Fig. 6 c-d** and described it in **lines 284 to 292**, and mitochondrial pyruvate supply, shown in **supplementary fig. 10 and described in lines 340 to 341**. Neither of these showed a rescue by MCU OE.

Additionally, we have performed citrin and MICU1 double KD as MICU1 KD is known to increase basal mitochondrial Ca²⁺ level, but the double KD also didn't rescue basal mitochondrial ATP level or ATP production of citrin KD cells shown in **Fig. 6 c-d** and described it in **lines 284 to 292**.

• The fact that perturbation of subcellular Ca²⁺ homeostasis does not affect the activity of the mitochondrial pyruvate dehydrogenase is not sufficient to conclude that Ca²⁺ perturbation does not affect mitochondrial dehydrogenases. Please modify this assumption throughout the manuscript.

We thank the reviewer for pointing to this subtle issue. We understand that indeed Ca²⁺ perturbation will affect mitochondrial Ca²⁺ sensitive dehydrogenases, but the main point we tried to relay to the readers is that we are using an acute perturbation of basal Ca²⁺ in a range of 20-40 nM, which, under our experimental conditions, is affecting citrin activity more than the activity of the

dehydrogenases. To support this claim, we have measured mitochondrial NADH production, which was reduced in 0Ca, but rescued by pyruvate supplementation, same as mitochondrial ATP production, which led us to the conclusion that TCA cycle dehydrogenases are not the main recipients/players during this slight and acute sub-cellular Ca^{2+} perturbation. Additionally, this assumption was supported by the fact that MCU KD didn't have such drastic effect on mitochondrial bioenergetics as did citrin KD (**Fig. 6**). We have modified the manuscript to make it clearer regarding this subtle point and refrained from making unjustified assumptions (e.g Title of section 2.2.).

Reviewer #3 (Remarks to the Author):

The work of Koshenov and colleagues has been designed to investigate principal regulating mechanisms of basal mitochondrial bioenergetics as fundamental processes for conditioning of cellular metabolic wiring. They found that Ca^{2+} fluxes from the ER-mitochondria contact sites activate the mitochondrial metabolism and keep mitochondria energized. Furthermore, they identified citrin as the primary regulator of this Ca^{2+} -dependent regulation. Finally, they affirm that manipulation of these Ca^{2+} dynamics may be useful to reprogram the mitochondrial metabolism in resting and pathological condition.

At today, it is widely accepted that intracellular Ca^{2+} variations, in particular modification of the Ca^{2+} transmission between ER and mitochondria, modulate the mitochondrial energy production with great impact with the overall cellular metabolism. In the first figures, authors focus their attention to this aspect. They give another demonstration of this, they establish a protocol that minimally perturbs basal subcellular Ca^{2+} homeostasis affecting the mitochondria and sub-mitochondrial compartments and reveal important aspects of both intracellular and extracellular Ca^{2+} in preserving the mitochondrial and cellular energetics.

However, in my opinion, I think that the primary finding of this work is to have identified citrin as the responsible for the metabolic rewiring during disturbed subcellular Ca^{2+} homeostasis. At demonstration the title of the manuscript is: "Citrin mediated metabolic rewiring in response to altered basal subcellular Ca^{2+} homeostasis".

Thus, I would encourage authors to perform new experiments aimed to demonstrate these findings. In particular authors should deep investigate about the importance of citrin in Ca^{2+} signaling. No experiments have been performed to demonstrate that citrin regulates Ca^{2+} at different compartments (cytosol, ER and mitochondria) as well as they lack to demonstrate that citrin alter the Ca^{2+} transmission between ER and mitochondria. I think these experiments are crucial since authors focus their attention to Ca^{2+} and observe that citrin deletion has similar effects like those observed in the condition with MCU deleted.

We thank the referee for these valuable and insightful comments. In order to address the referees point, we have assessed mitochondrial Ca^{2+} uptake under citrin KD condition. We can confirm that citrin KD doesn't affect basal matrix Ca^{2+} levels or mitochondrial Ca^{2+} uptake upon stimulation with IP_3 generating agonist. The results are shown in **Supplementary Fig. 9**.

Furthermore, authors should include more evidences of the alteration of the mitochondrial energetic in the citrin and MCU KD conditions by performing experiments like those reported in Figure 2.

We thank the referee for this helpful comment. Throughout the manuscript we have attempted to elucidate regulation mechanisms of basal mitochondrial bioenergetics. We were able to

demonstrate that IMS Ca^{2+} regulated citrin activity is important for that regulation and that intracellular calcium imbalance can trigger metabolic rewiring in citrin dependent manner. To show this, we have already performed the experiments mentioned by the referee (similar to those shown in figure 2) for citrin and MCU KD (**Figure 6**). Additionally, we have added mitochondrial pyruvate supply (**Fig. 7 c-e** and described in **lines 332-341**) and seahorse data for citrin and MCU KD (**Supplementary fig. 8**, described in **lines 276-283**), which confirmed that citrin plays an important role in the regulation of basal metabolism, as citrin KD resulted in reduced basal, and ATP production linked respiration, increased ECAR, and reduced mitochondrial pyruvate supply.

Authors identify Citrin a responsible for cytosolic and mitochondrial metabolic rewiring. I think they should also explore this possibility in pathologic context. Does exist a particular disease in which citrin is downregulated or over-expressed? Is it possible to counteract the disease phenotype by modulating the citrin expression?

We thank the referee for this comment. We have added additional information in discussion on cancer cells up-regulating citrin levels as well as type 2 citrullinemia, **lines 596-503**.

Finally, in figure 6 authors demonstrate that citrin controls pyruvate availability for mitochondria. In the subsequent figure authors demonstrate that mitochondrial bioenergetics are influenced by basal MAM Ca^{2+} flux. Here, no experiments have been conducted with citrin. These findings seem to be unrelated to each other.

We thank the referee for this insightful comment. As we have demonstrated that intracellular calcium imbalance is affecting bioenergetics and triggering metabolic rewiring in citrin dependent manner, we wanted to clarify which Ca^{2+} flux is important for its regulation. As citrin is localized at mitochondrial IMS and there is no apparent threshold for calcium uptake into IMS, we wanted to distinguish between the importances of total cytosolic vs. MAM calcium flux for citrin dependent regulation of mitochondrial bioenergetics (old **Figure 7**, now **figure 8**). We have further elaborated this point as requested by reviewer 1 by performing KD experiments with IP3R2 and Orai1 (new **Figure 8 d-e**), which supports our claim that basal MAM Ca^{2+} flux is important for IMS Ca^{2+} regulation of citrin/mitochondrial activity.

Minor points:

- Please simplify the schematic illustration of Figure 8.

We thank the referee for this comment. We tried to summarize and simplify the main points of the manuscript in the form of figure 8 (now figure 9). We are afraid that further simplification will make the figure lose its purpose. We hope for the referee's understanding

- transient knockdown of citrin should be also demonstrated by immunoblot

We have added WB of citrin KD as **Supplementary Fig. 5**

- Citrin KD profoundly reduces the energetic production. Authors should verify if this condition also affects the cell growth and cell death processes like apoptosis.

We thank the referee for this suggestion. We have added results of cell viability and apoptosis assays as **Supplementary Fig. 6**, procedure described in **lines 743-751**.

REVIEWERS' COMMENTS:

Reviewer #1 (Remarks to the Author):

The authors have satisfactorily addressed my comments by performing new experiments and by giving clarifications.

Reviewer #2 (Remarks to the Author):

The authors adequately responded to my concerns. I would recommend moving the blot of total PDH (now in supply. fig. 3) to figure 3, in correspondence to the related phospho-PDH blot. In addition, there is no reference in the main text to panel 1D. Figure 1a now misses i, ii, iii, iv.

Reviewer #3 (Remarks to the Author):

Dear Authors,

I would like to thank you for having followed my suggestion and considered my concerns. In my personal opinion, you have well answered to all my questions. For this, the manuscript has been significantly improved.

I would only ask you to make a little modifications of your manuscript.

In the previous revision round, I asked to simplify your schematic illustration. You (rightly) wrote that: "We are afraid that further simplification will make the figure lose its purpose. We hope for the referee's understanding".

I agree with your decision. I only ask you to extend the figure caption. By this way, a reader can easily understand the schematic illustration.

Once you address this, the manuscript will be ready to be published

Reviewer #1 (Remarks to the Author):

The authors have satisfactorily addressed my comments by performing new experiments and by giving clarifications.

Reviewer #2 (Remarks to the Author):

The authors adequately responded to my concerns. I would recommend moving the blot of total PDH (now in supply. fig. 3) to figure 3, in correspondence to the related phospho-PDH blot.

The total PDH blot has been moved to figure 3

In addition, there is no reference in the main text to panel 1D.
Figure 1a now misses i, ii, iii, iv.

The panels have been corrected

Reviewer #3 (Remarks to the Author):

Dear Authors,

I would like to thank you for having followed my suggestion and considered my concerns.
In my personal opinion, you have well answered to all my questions.
For this, the manuscript has been significantly improved.

I would only ask you to make a little modifications of your manuscript.

In the previous revision round, I asked to simplify your schematic illustration. You (rightly) wrote that: "We are afraid that further simplification will make the figure lose its purpose. We hope for the referee's understanding".

I agree with your decision. I only ask you to extend the figure caption. By this way, a reader can easily understand the schematic illustration.

Once you address this, the manuscript will be ready to be published

Figure caption on fig. 9 has been extended for more clarity

From the editor:

In particular, we ask that you address the following:

- Provide the source data underlying the graphs and charts in the main manuscript (excel) as a Supplementary Data file and mention this in the Data Availability statement (done)
- Provide all addgene IDs in the Data Availability statement (done)
- Rename and rephrase the Competing Interests statement (done)
- Rename "Supplementary figures" to "Supplementary Information" (done)
- For all graphs and charts depicting a mean with error bars, individual data points must be shown (done)
- All microscopy images require scale bars (Figure 1 b can't have a scale bar as it is a 3D reconstruction of image stacks, the figure legends has been changed accordingly).
- Display molecular weight size markers for all gels/blots and provide the full unprocessed images as a Supplementary Figure (done)
- Avoid use of "as previously described" and detail the methods in full (done)
- Provide dilutions of antibodies used (done)
- Expand relevant section to "Statistics and Reproducibility" (done)
- Ensure all affiliations have cities and countries in addresses (done)
- Avoid use of speech marks and italics for emphasis (done)